# Molecular dynamics simulation based prediction of T-cell epitopes for the production of effector molecules for liver cancer immunotherapy

Sidra Zafar[1], Yuhe Bai[2], Syed Aun Muhammad[1]*, Jinlei Guo[3], Haris Khurram[4,5], Saba Zafar[6], Iraj Muqaddas[1], Rehan Sadiq Shaikh[7], Baogang Bai[8,9,10]*

1 Institute of Molecular Biology and Biotechnology, Bahauddin Zakariya University Multan, Multan, Punjab, Pakistan, 2 Department of Computer Science, Sorbonne University, Paris, France, 3 School of Intelligent Medical Engineering, Sanquan College of Xinxiang Medical University, Xinxiang, Henan, China, 4 Department of Mathematics and Computer Science, Faculty of Science and Technology, Prince of Songkla University, Pattani Campus, Pattani, Thailand, 5 Department of Sciences and Humanities, National University of Computer and Emerging Sciences, Chiniot, Punjab, Pakistan, 6 Department of Biochemistry and Biotechnology, The Women University Multan, Multan, Punjab, Pakistan, 7 Center for Applied Molecular Biology (CAMB), University of the Punjab, Lahore, Punjab, Pakistan, 8 School of Information and Technology, Wenzhou Business College, Wenzhou, Zhejiang, China, 9 Zhejiang Province Engineering Research Center of Intelligent Medicine, Wenzhou, China, 10 The 1st School of Medical, School of Information and Engineering, The 1st Affiliated Hospital of Wenzhou Medical University, Wenzhou, Zhejiang, China

* aunmuhammad78@yahoo.com (SAM); 20190165@wzbc.edu.cn (BB)

**Data Availability Statement:** All relevant data are within the paper and its Supporting information files.

## Abstract

Liver cancer is the sixth most frequent malignancy and the fourth major cause of deaths worldwide. The current treatments are only effective in early stages of cancer. To overcome the therapeutic challenges and exploration of immunotherapeutic options, broad spectral therapeutic vaccines could have significant impact. Based on immunoinformatic and integrated machine learning tools, we predicted the potential therapeutic vaccine candidates of liver cancer. In this study, machine learning and MD simulation-based approach are effectively used to design T-cell epitopes that aid the immune system against liver cancer. Antigenicity, molecular weight, subcellular localization and expression site predictions were used to shortlist liver cancer associated proteins including *AMBP*, *CFB*, *CDHR5*, *VTN*, *APOBR*, *AFP*, *SERPINA1* and *APOE*. We predicted CD8+ T-cell epitopes of these proteins containing LGEGATEAE, LLYIGKDRK, EDIGTEADV, QVDAAMAGR, HLEARKKSK, HLCIRHEMT, LKLSKAVHK, EQGRVRAAT and CD4+ T-cell epitopes of VLGEGATEA, WVTKQLNEI, VEEDTKVNS, FTRINCQGK, WGILGREEA, LQDGEKIMS, VKFNKPFVF, VRAATVGSL. We observed the substantial physicochemical properties of these epitopes with a significant binding affinity with MHC molecules. A polyvalent construct of these epitopes was designed using suitable linkers and adjuvant indicated significant binding energy (>-10.5 kcal/mol) with MHC class-I and II molecule. Based on *in silico* cloning, we found the considerable compatibility of this polyvalent construct with the *E. coli* expression system and the efficiency of its translation in host. The system-level and machine learning based

**Funding:** The author(s) received no specific funding for this work.

**Competing interests:** NO authors have competing interests.

cross validations showed the possible effect of these T-cell epitopes as potential vaccine candidates for the treatment of liver cancer.

## 1. Introduction

Liver cancer (LC) is one of the major cause of deaths throughout the world. It is a progressive, multifactorial and prevalent disease accounts for almost 1 million cases around the globe recorded in 2020 [1] indicating nearly 90% occurrence of all liver cancer [2]. Unfortunately, it has been observed that LC is extremely non-effective to standard cytotoxic chemotherapies [3]. Even today, due to less understanding of pathophysiological mechanism, classical diagnostic and therapeutic options, we are facing challenges to cure liver cancer [4]. To overcome this situation, we need to develop more safer, effective and targeted therapeutic strategies. Recent studies found a sub-class of cells with stem cell properties that are involved in the initiation of the tumor makes the tumor persistence, ability to relapse, metastasis, radio resistance and chemoresistance [5]. This causes the tumor cells to withstand with both traditional and innovative therapeutic regimens like Sorafenib [6].

It has been reported that adaptive immunity cannot be effective enough to kill cancer cells, leading the cancer cells to become resistant to the immune system. Initially, the immune system brings about a strong immune response. This response causes some cancer cells to eradicate, while others with low immunogenicity escape detection and proliferate [7]. Macrophages, dendritic cells, natural killer (NK) cells are key innate immune effector molecules in cancer patients [8]. Tumor cells are modified cells always attempting to evade the immune system, which results in a reduction in the expression of antigen-presenting receptors of the Major Histocompatibility Complex class I (MHC-I). Due to the lack of self-recognition, these MHC-I knocked-down cells fell prey to NK cells, which kill them [9]. Dendritic cells serve as a link between the innate and adaptive immune systems. Distinctive to innate immunity, which responds immediately, adaptive immunity takes a few days to respond [10]. T cells only recognize and digest these antigens when they are dispensed by specialized innate immune cells [11]. B cells are more notable because they secrete antibodies (humoral immunity) that exert anticancer functions through activating complement, increasing cellular cytotoxicity, and arbitrating cancer cell phagocytosis [12].

One of the immunotherapeutic approaches to cure cancer is using therapeutic cancer vaccines [13]. The aim behind availing these vaccines for the treatment of LC is to identify tumor associated antigens (TAA) and tumor specific antigens (TSA) as potential vaccine candidates [11]. Computational approaches are utilized to predict and map new vaccine peptides using *in-silico* analysis [14]. In this regard, epitope-based vaccines have been made possible by improvements in epitope prediction for the preparation of these preparations [15]. Compared to conventional vaccines, epitope-based vaccines provide a number of benefits, such as the capacity to target certain antigens and the potential to elicit a higher immune response. Epitopes are specifically functional in preventing diseases, developing vaccines, detecting and treating diseases that is why clinical and biological researchers are interested in them [16]. To develop a significant liver cancer vaccine, scientists are now employing computational methods to anticipate B and T cell epitopes.

The recent approaches involve the expertise in the field of bioinformatics, genetics, and immunology. Instead of isolation, culturing and studying the cells and tissues, researchers are identifying promising vaccine candidates by computational techniques and data mining. Utilizing only

specific antigenic proteins, these techniques not only save time and resources but has the potential to produce safer vaccines. Here are some already approved reverse vaccinology-based vaccines include COVID-19 vaccine against virus [17], Human papillomavirus (HPV) vaccine (Cervarix, Gardasil) [18], Shingles vaccine (Shingrix) for the treatment of secondary varicella infection [19], Meningococcal B vaccine to provide the protection against bacterial meningitis [20].

T-cell epitopes are more significant in developing immunotherapeutic vaccines as they have the potential to provoke other components of immune system. These vaccines are reported more effective against various cancers in pre-clinical and clinical trials [21]. To achieve broad spectral therapeutic effect, preferably T-cell epitopes are designed from conserved regions of tumor associated or tumor specific proteins, as cancer cells find it difficult to escape T-cell mediated immunity [22]. It has been observed that T-lymphocytes provide more robust cell mediated immunity compared to B-lymphocytes that are capable of producing humoral immunity [22, 23].

In this study, we used consolidative and system-level framework to identify the potential liver cancer vaccine candidates involving data mining, antigenicity predictions, genomic conservations, protein-protein interaction studies, molecular interaction and binding affinity analysis, and machine learning and MD simulation based cross-validation techniques. Prediction of T- cell epitopes for the production of effector molecules to cure liver cancer opens the door for the development of novel and efficient therapeutic vaccine that may greatly improve the prognosis and treatment options (Fig 1).

## 2. Materials and methods

### 2.1 Ethical statement

There is no human and animal involvement. The ethical statement is not applicable to this study. Our system level framework involves a variety of immunoinformatic and machine learning based analysis, data mining, software and tools to predict the T-cell epitopes of liver cancer (S1 Table).

### 2.2 Retrieval and screening of proteins

Human Protein Atlas (HPA) [24] (https://www.proteinatlas.org/), UniProt (https://www.uniprot.org/), NCBI (https://www.ncbi.nlm.nih.gov/), and Liverome databases [25] (http://liverome.kobic.re.kr/) were used to find liver cancer associated proteins. The FASTA format of these protein sequences were retrieved followed by the screening based on antigenicity, molecular weight, and subcellular localization. The molecular weight was determined by the ExPASY server [26] (https://web.expasy.org/compute_pi/) and the antigenicity was predicted using the VaxiJen tool [27] (http://www.ddg-pharmfac.net/vaxijen/). We determined the Sub cellular localization of these proteins was determined by CELLO (http://cello.life.nctu.edu.tw/) and UniProt servers [28]. To predict and screen the proteins based on their optimal expression sites, preferably liver tissues, a GenomicScape server was applied [29] (http://genomicscape.com/).

### 2.3 Identifying tumor-associated and tumor-specific proteins

Selected proteins were further categorized as tumor-specific antigens (TSA) and tumor-associated antigens (TAA). TAA were located on all normal and cancerous cells of the body but over expressed in tumor cells while TSA are located on cancerous cells only and are specific to these cells. We used TRON Cell Line Portal [30] (http://celllines.tron-mainz.de/) and Tumor-Specific Neo-Antigen Database (TSNAsb) (http://biopharm.zju.edu.cn/tsnadb/browse/) to identify the TAAs and TSAs [31]. These servers comprehensively elaborate the neoantigens indicating specific tumor antigen and a potential target of a particular type of cancer immunotherapy.

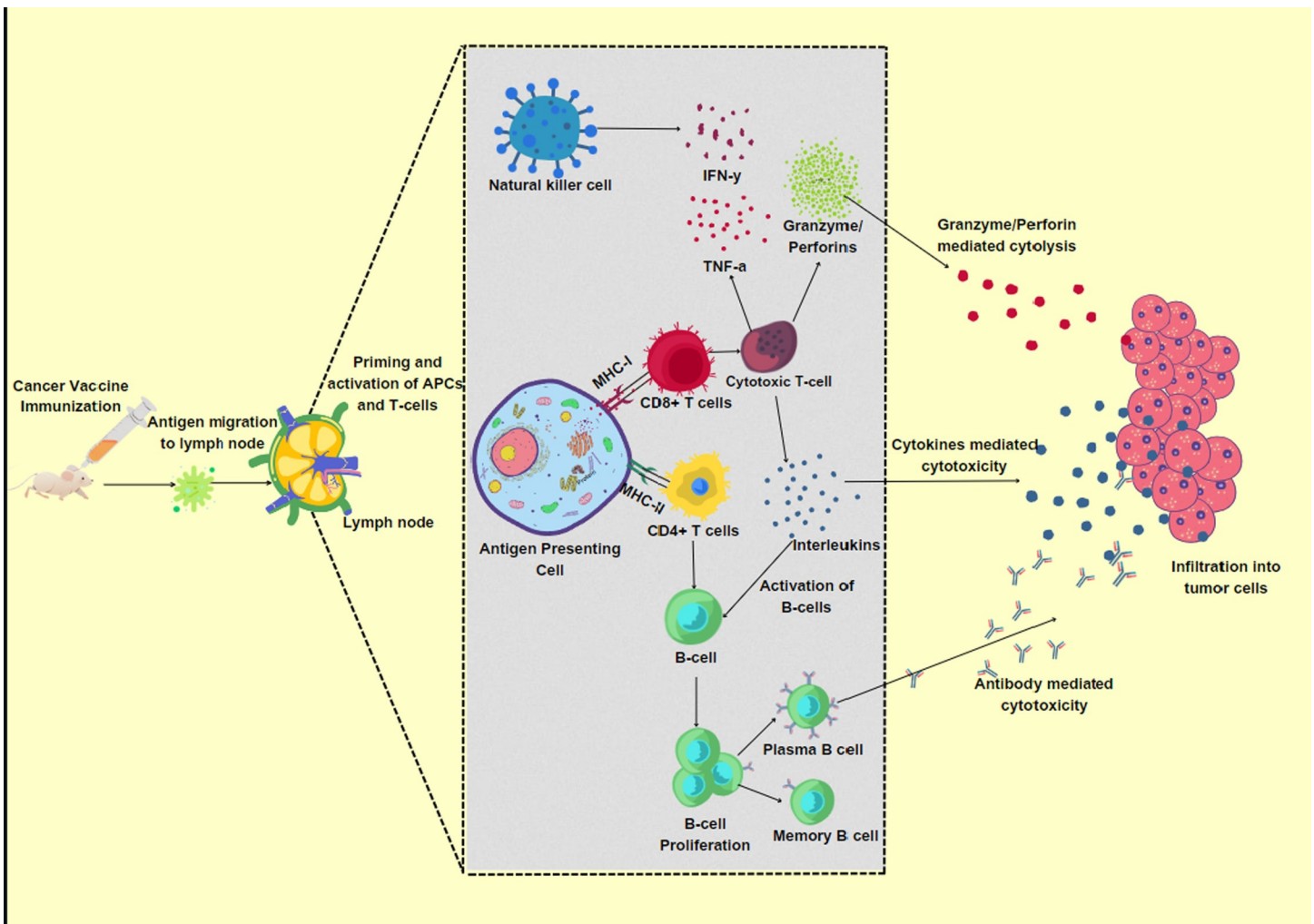

**Fig 1. Schematic representation indicating the hypothesis of our study.** T-cell epitopes provoke and enhance the action of immune response to eradicate the tumor cells.

## 2.4 Protein-Protein Interaction (PPI) and network analysis

The purpose of the PPI network is to overview the interactions between the target proteins and the host proteins presenting the overview of the directly and indirectly associated proteins in cancer. Human Annotated Protein-Protein Interactions (HAPPI) database (http://discovery. informatics.uab.edu/HAPPI/) was used to find all the proteins that interact with our target (selected) protein [32]. The network was constructed by using Cytoscape (https://cytoscape. org/) software version 3.10.1. In this network, nodes represented the proteins and edges showed the interaction [33]. Cancer associated and TSAs/TAAs nodes were differentiated and highlighted by different colors.

## 2.5 Prediction of T cell epitopes, conservation and antigenicity analysis

The software HLAPred (http://crdd.osdd.net/raghava/hlapred/) was considered for designing T-cell epitopes [34]. FASTA sequences were uploaded to predict MHC Class I and II epitopes, population coverage and multiallelic epitopes were analyzed. The UCSC Xena server [35] (https://xena.ucsc.edu/) was employed for epitope conservation to analyze the broad spectral

therapeutic effect of these proteins. Epitopes were screened on the basis of antigenicity, immunogenicity [26] by using the VaxiJen server [27] and Immune Epitope Database (IEDB) [36] (https://www.iedb.org/) respectively.

## 2.6 Development of polyvalent construct

To determine the sequence of epitopes in the final polyvalent construct, all of the screened epitopes were analyzed to find the best compatible combination [37]. Linkers including AAY and GPGPG were used for MHC class-I and II constructs respectively [38, 39]. Initially, two epitopes were linked by the selected likers to build the 3D model using Chimera, followed by the cross-validation using HADDDOCK 'GURU' interface, refinement and C-port modules [40]. While developing polyvalent construct, clusters of double epitopes were linked with the third and fourth epitopes followed by the same cross-validation analysis to find the best affinity models.

## 2.7 Safety profile assessment

Safety profile parameters of epitopes and polyvalent construct were examined including stability, SVM score (toxicity) and allergenicity of the conserved epitopes [41]. Support Vector Machine (SVM) model of amino acid composition of epitopes was used to predict the toxicity through online servers ToxinPred and PredSTP [42] (http://crdd.osdd.net/raghava/toxinpred). Stability of a peptide in test tube or in synthesized drug form is indicated by the value of instability index. Predicting allergenicity helps ensure vaccine safety. Epitope's allergenicity was predicted by AllerTop v. 2.0 [43, 44] (https://www.ddg-pharmfac.net/AllerTOP/method.html).

## 2.8 Physicochemical properties analysis

The physicochemical properties of epitopes were analyzed to observe hydropathicity (GRAVY), charge, theoretical pI, half-life, aliphatic index, molecular weight and solubility [45]. A thorough understanding of the hydropathicity of each individual amino acid help to anticipate the overall three-dimensional structure of a protein from its amino acid sequence. ExPASy and Peptide 2.0 were utilized to forecast the hydropathicity of peptides [26]. For a protein to be soluble in water it must have charge on it and for this purpose, we used ProtParam (https://web.expasy.org/protparam/) to determine the charge, theoretical pI and aliphatic index. Half-life of peptide is different in different species, we determined the half-life of peptides in human using PEPlife, ToxinPred and ExPASy. Similarly, the molecular weight was determined by the ExPASY server [26] CamSol (http://www-vendruscolo.ch.cam.ac.uk/camsolmethod.html) was used to analyze the solubility of individual epitopes [46] while solubility of polyvalent construct was determined by the Protein-sol server [47] (https://protein-sol.manchester.ac.uk/). The immunodominant potential vaccine target against *Brucella melitensis* (C1) was used as positive control to compare the characteristics of designed polyvalent construct [48].

## 2.9 Proteasomal cleavage analysis

The processing and presentation of antigens via MHC represent a highly complex mechanism of proteasomal cleavage. A critical step in this mechanism is the recognition of specific antigen sequences for presentation to immune cells [49]. TAPs (Transporters associated with antigen processing) were also involved in antigen presentation to MHC after cleavage with

proteasomes. We have carried out this analysis by NetChop (IEDB) server aiding in identifying potential sites within protein sequences susceptible to proteasomal cleavage [50].

## 2.10 In silico cytokine release evaluation

To analyze the potential immunogenicity and efficacy of our epitopes, we predicted the cytokine release (IFN-γ, IL-4, IL-6, and IL-10) from the activated T-lymphocytes using IFNepitope [51], IL-4Pred [52] (http://crdd.osdd.net/raghava/il4pred/), IL-6Pred [53] (https://webs.iiitd.edu.in/raghava/il6pred/), and IL-10Pred [54] (http://crdd.osdd.net/raghava/IL-10pred/) servers respectively. These interleukins play a complex and multifaceted role in liver cancer diagnosis and treatment. Th2 cells are a particular subset of immune cells that can be activated by IL-4. IL-6 has the capacity to trigger apoptosis and can stimulate $CD^{8+}$ T cells along with other immune cells that target cancer directly [55]. IL-10 can strengthen immune responses against malignancies by encouraging the development of tumor-specific T lymphocytes [56].

## 2.11 3D Modelling and quality assessment

Robetta (https://robetta.bakerlab.org/) and I-Tesser (https://zhanggroup.org/I-TASSER/) tools were used to construct the 3D illustrations of the whole protein [57]. The sequence of the epitope was highlighted in its native protein model after it was processed in ChimeraX [58] (https://www.cgl.ucsf.edu/chimerax/). The 3D models of epitopes and polyvalent construct were generated using PEP-FOLD server (https://bioserv.rpbs.univ-paris-diderot.fr/services/PEP-FOLD/) and I-Tesser [59, 60] as it is an essential step to predict the binding energy.

The quality of these models were assessed by the ERRAT [61] (https://saves.mbi.ucla.edu/) and MolProbity [62] tools. The PEP-FOLD server presented the QMEAN (Qualitative Model Energy Analysis) scores highlighting the homology and similarity of predicted models with experimental structures [60]. Similarly, Ramachandran plots [63] explained the total number of amino acids available in favorable and unfavorable regions. *In silico* bonding between the chains was checked by performing disulfide engineering (http://cptweb.cpt.wayne.edu/DbD2/index.php) to maintain and improve the stability of construct. Introduction of disulfide bonds also held the protein in specific 3D conformation to become more immunogenic by exposing the certain immune stimulating regions [48].

## 2.12 Molecular docking of epitopes and polyvalent construct

The binding energies of the selected epitopes were investigated by molecular docking using Molecular Operating Environment (MOE) software [64]. The lesser the binding energy, greater the affinity of the ligand (epitopes) with MHC class-I (PDB ID: 1W72) and MHC class-II (PDB ID: IBX2) as potential targets [65]. This binding interprets the appropriate orientation and attachment of the ligands to the target in order to produce a stable complex [66]. The epitopes of MHC-I and MHC-II were checked for their affinity with their respective receptors included MHC class-I receptor (*HLA-A1*) with PDB ID: 1W72 [67] and MHC class-II receptor (*HLA-DR2*) having PDB ID: 1BX2 [68].

ClusPro (https://cluspro.org/help.php) server was utilized to dock the polyvalent construct with their target molecules of class MHC class-I and II [69]. Additionally, the interaction of ligand-target complex were visualized by LigPlot+ software [70].

## 2.13 Molecular dynamics simulation of construct

The structure of the polyvalent construct was stabilized by GROMACS, an updated MD modeling tool [71] that provides a comprehensive platform for preparation, analysis, and

calculations. In order to maintain the stability, a number of parameters were operated including velocity, pressure, temperature, energy minimization, RMSD, RMSF, and Rg (radius of gyration). Optimized Potential for Liquid Simulation force-field was selected and the protein was controlled in a rhombic dodecahedron cubic box to accommodate solvent molecules. Solvation was performed by SPCE water model followed by electro-neutralization and energy minimization under steepest descend algorithm. Program *GMX* energy was applied to observe some or all of the energies and other parameters like density, box-volume, box-sizes, pressure tensor, and total pressure. One can select from a list of set of energies, such as potential, kinetic, or total energy, or from a list of individual energies, such as dihedral or Lennard-Jones energies. The center of mass velocity is defined as:

$$V_{com} = \frac{1}{M} \sum_{i=1}^{N} m_i V_i$$

For the real-time monitoring of the system's overall mass, $M = \sum_{i=1}^{N} m_i$. However, it is directed to eliminate the center-of-mass velocity with each step.

The temperature is given by the total kinetic energy of the N-particle system:

$$\boldsymbol{E}_{kin} = \frac{1}{2} \sum_{i=1}^{N} m_i V_i^2$$

Protein temperature was stabilized using NVT isothermal-isochoric ensemble equilibration at 100 ps, up to a specified value. The quantities we want to compute were obtained from a constant temperature (NVT) ensemble, also known as the canonical ensemble, although the direct use of molecular dynamics also includes the NVE (constant number, constant volume, constant energy ensemble). To simulate constant temperature, GROMACS can employ the Berendsen weak-coupling method [72], the extended ensemble Nosé-Hoover scheme [73, 74], the Andersen thermostat [75], or a velocity-rescaling scheme [76]. The Berendsen algorithm simulates a weak coupling with first-order kinetics to an external heat bath with a fixed temperature $T_0$. This algorithm has the effect on the maintenance of system temperature from $T_0$ in accordance with:

$$\frac{DT}{dT} = \frac{T_o - T}{\tau}$$

This indicates that the temperature deviation decays exponentially with a given time constant $\tau$. The benefit of this coupling technique is that the coupling strength can be adjusted to the required specifications.

Difference between (Ekin) kinetic energy and virial ($\Xi$) provide the pressure tensor P.

$$P = \frac{2}{v}(E_{kin} - \Xi)$$

At every nPC steps, the Berendsen method rescales the coordinates and box vectors with a matrix μ. This results in a first-order kinetic relaxation of the pressure towards a specified reference pressure P0, as:

$$\frac{dP}{dt} = \frac{P_0 - P}{\tau p}$$

Pressure and temperature were maintained at 1 bar and 310 k, respectively.

Energy minimization (EM) was done at the 5000 steps in order to get the final energy minimized structure. Conjugate gradients, steepest descent and l-bfgs (limited-memory Broyden-Fletcher-Goldfarb-Shanno quasi-Newtonian minimizer) are three methods for minimizing energy in GROMACS. EM is merely one of the *mdrun* program's options. Steepest descent algorithm was used as it is simple and robust method. The vector of all 3N coordinate is referred to as the vector r. It is necessary to first specify a maximum displacement $h0$ (e.g., 0.01 nm). Potential energy and force F were computed first. The calculation of new positions was done by:

$$r_n + 1 = r_n + \frac{F_n}{\max(|F_n|)} h_n$$

Where 'Fn' was the force, or the negative gradient of the potential V, and n is the maximum displacement. The largest scalar force on any atom is indicated by the notation 'max(|Fn|)'.

Radius of gyration (Rg) was used to measure the compactness of structure was measured, Rg is calculated by:

$$\boldsymbol{R}_g = \left( \frac{\sum_i \|r_i\|^2 m_i}{\sum_i m_i} \right)^{\frac{1}{2}}$$

where ri is position of atom i with respect to the molecule's center of mass and mi is the mass of atom i. This is very helpful in characterizing the polymers of solutions and proteins. The software will also yield the radius of gyration around the coordinate axis (or, alternatively, primary axes) by simply adding the radii components orthogonal to each axis, for example,

$$R_{g,x} = \left( \frac{\sum_i \left( r_{i,y}^2 + r_{i,z}^2 \right) m_i}{\sum_i m_i} \right)^{\frac{1}{2}}$$

The program *GMX* rms can be used to compute the root mean square deviation (RMSD) of specific atoms in a molecule with respect to a reference structure by least-square fitting the structure to the reference structure ($t_2 = 0$) and then computing the RMSD.

$$RMSD(t_1, t_2) = \left[ \frac{1}{N} \sum_{i=1}^{N} \|r_i(t_1) - r_i(t_2)\|^2 \right]^{\frac{1}{2}}$$

Where position of atom 'i' at time 't' is indicated by ri(t). It should be noted that fitting does not always require the use of the same atoms for the RMSD calculation. For example, a protein is often fitted on its backbone atoms (N, Cα, and C), but its RMSD can be calculated of the protein as a whole or of the backbone.

Alternatively, one can calculate RMSD ($t_1, t_2$) using a structure at time $t_2 = t_1 - \tau$, instead of comparing the structures to the initial structure at time t = 0 (such as a crystal structure). This provides some understanding of the mobility in relation to τ. Another way to create a good graphical representation of a trajectory is to create a matrix using the RMSD as a function of t1 and t2. Transitions in a trajectory will be evident in such a matrix if any exist. Fit-free method program was also run to calculate RMSD as:

$$RMSD(t) = \left[ \frac{1}{N^2} \sum_{i-1}^{N} \sum_{j=1}^{N} \|r_{ij}(t) - r_{ij}(0)\|^2 \right]^{\frac{1}{2}}$$

where the distance 'rij' between atoms at time 't' is compared with the distance between the same atoms at time 0.

The stabilized vaccine construct was subjected to temperature, pressure, and density stabilization using an NPT ensemble with 50,000 steps total for the entire procedure. For the equilibrated and stabilized construct, an MD simulation with 50,000 steps was performed at 50 ns.

## 2.14 Immune simulation analysis of construct

An immune simulation of a polyvalent construct's sequence in the form of its amino acid sequence is performed using the C-IMMSIM (https://kraken.iac.rm.cnr.it/C-IMMSIM/index. php) server. The humoral and cellular responses of a mammalian immune system in response to the vaccine design are described by this web server.

In conclusion, C-IMMSIM presents images that include representations of the major cell classes of the myeloid lineage (macrophages (M) and dendritic cells) as well as the lymphoid lineage (T helper lymphocytes (Th), cytotoxic T lymphocytes (CTL), B lymphocytes, and antibody-producer plasma cells, PLB) [77]. The initial population of lymphocytes is determined by parameters in C-IMMSIM, including simulation volume, host haplotype selection, injection schedule (both time and dose), and random seed. In order to generate an effective and long-lasting immune response, the study's simulated parameters included a vaccine devoid of lipopolysaccharides (LPS) and number of Antigen (Ag) to inject were set at 1000, three vaccine doses were set at time intervals of 1,63 and 126-time steps (1 time step is equal to 8 hours) then simulation volumes and steps were set to 10 and 1050, respectively. The other perimeters were kept as default.

## 2.15 In silico codon optimization and cloning of construct

In *E. coli (strain K12)*, the expression level of the multi-epitope vaccine was measured using the Java Codon Adaptation Tool (J.Cat server: (http://www.prodoric.de/JCat). The reverse translational analysis was performed using this server. The output is cDNA sequence, which was further examined for codon optimization to get a compatible sequence with prokaryotic expression vector. To get the maximum expression outcomes, J.Cat provide the GC content and codon adaptation index (CAI) value [78]. Finally, polyvalent construct was then cloned using SnapGene software (version 5.2.3) into the pET-28a (+) plasmid [79]. For restriction and cloning, Hind III and Bam HI restriction sites were incorporated at the polyvalent construct's N and C terminals, respectively.

## 3. Results

### 3.1 Protein retrieval and screening

We accessed 546 liver cancer associated proteins from the databases and from this total, 125 proteins are overexpressed in liver cancer patients. So, these cancerous proteins were selected for further screening and their potential ability of antigenicity. Off these 125 proteins, we predicted 21 are extracellular, 33 are membrane-bound, and 71 are intracellular proteins (Fig 2).

We found 15 antigenic proteins from the total sum of extracellular and membrane bound proteins based on the significant cut off parameters of VaxiGen database (>0.5). As larger molecular weight is associated with more immunogenicity [36], therefore, we analyzed the molecular weight of these 15 proteins and based on this parameter (>30kDa), ten proteins were retained (Table 1). The differential expression sites of these 10 proteins in different body tissues were investigated and finally 8 proteins were selected based on their significant expression in liver. These proteins includes CDHR5 (Cadherin-related family member 5), AMBP (Alpha-1-microglobulin/bikunin precursor), CFB (Complement factor B), VTN (Vitronectin), APOBR (Apolipoprotein B receptor), FETA (Alpha-fetoprotein), A1AT (Alpha 1-antitrypsin) and APOE (Apolipoprotein E) (Fig 3).

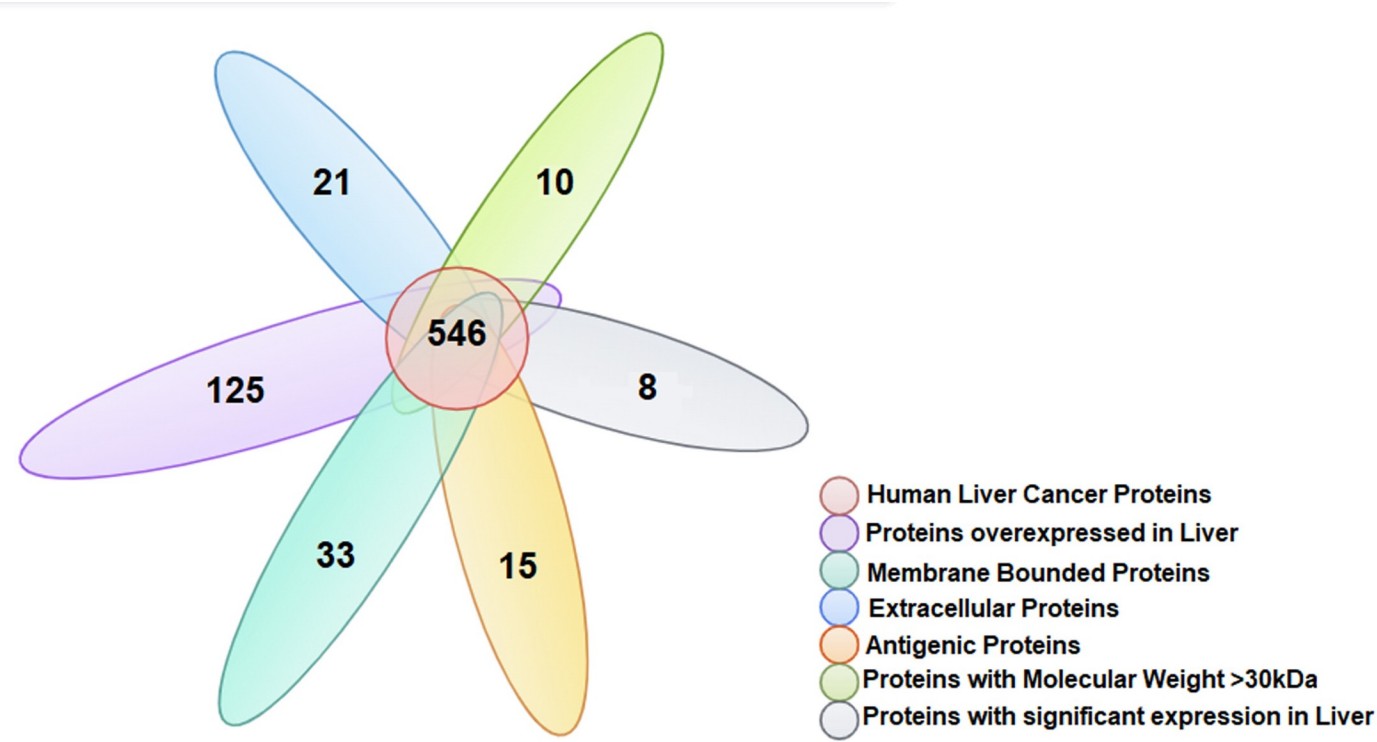

**Fig 2. Venn diagram shows the screening of retrieved liver cancer proteins for the prediction of T-cell epitopes.**

## 3.2 Prediction of tumor-specific and tumor-associated proteins

We predicted the tumor specific antigens (TSAs) and tumor associated antigens (TAAs) from the list of 8 selected proteins. Off eight, 5 proteins including VTN, CDHR5, APOBR, AFP and APOE are TSAs, while the remaining CFB, AMBP and SERPINA proteins are TAAs (Table 1). During data mining of liver cancer patients, we observed mutations in selected TSAs. For example, CDHR5 protein possessed the mutation at position 107 and indicated the substitution of glutamine with histidine. The other TSA protein VTN was mutated at the 253 positions, presenting isoleucine is replaced with asparagine. Similarly, in case of APOBR and AFP antigens, we analyzed mutation at 435 and 57 position respectively where aspartic acid is changed into glycine and glutamic acid is replaced with pyrrolysine.

**Table 1. Screening parameters of retrieved liver cancer proteins.**

| Sr. No. | Protein Name | Gene Name | Uniprot ID | TAA/TSA | Subcellular Localization | Antigenicity | Molecular weight (kDa) |
|---|---|---|---|---|---|---|---|
| 1 | Alpha 1 microglobulin/bikunin precursor | AMBP | P02760 | TAA | Extracellular | 0.519 | 38.99 |
| 2 | Complement factor B | CFB | P00751 | TAA | Extracellular | 0.61 | 85.562 |
| 3 | Cadherin 5 | CDHR5 | Q9HBB8 | TSA | Extracellular | 0.54 | 87.63 |
| 4 | Vitronectin | VTN | P04004 | TSA | Extracellular | 0.58 | 54.335 |
| 5 | Apolipoprotein B receptor | APOBR | Q0VD83 | TSA | Membrane Bounded | 0.84 | 115.6338 |
| 6 | Alpha-fetoprotein | AFP | P02771 | TSA | Extracellular | 0.86 | 68.67757 |
| 7 | Alpha-1-antitrypsin | SERPINA1 | P01009 | TAA | Extracellular | 0.73 | 46.73655 |
| 8 | Apolipoprotein E | APOE | P02649 | TSA | Extracellular | 0.66 | 36.15408 |

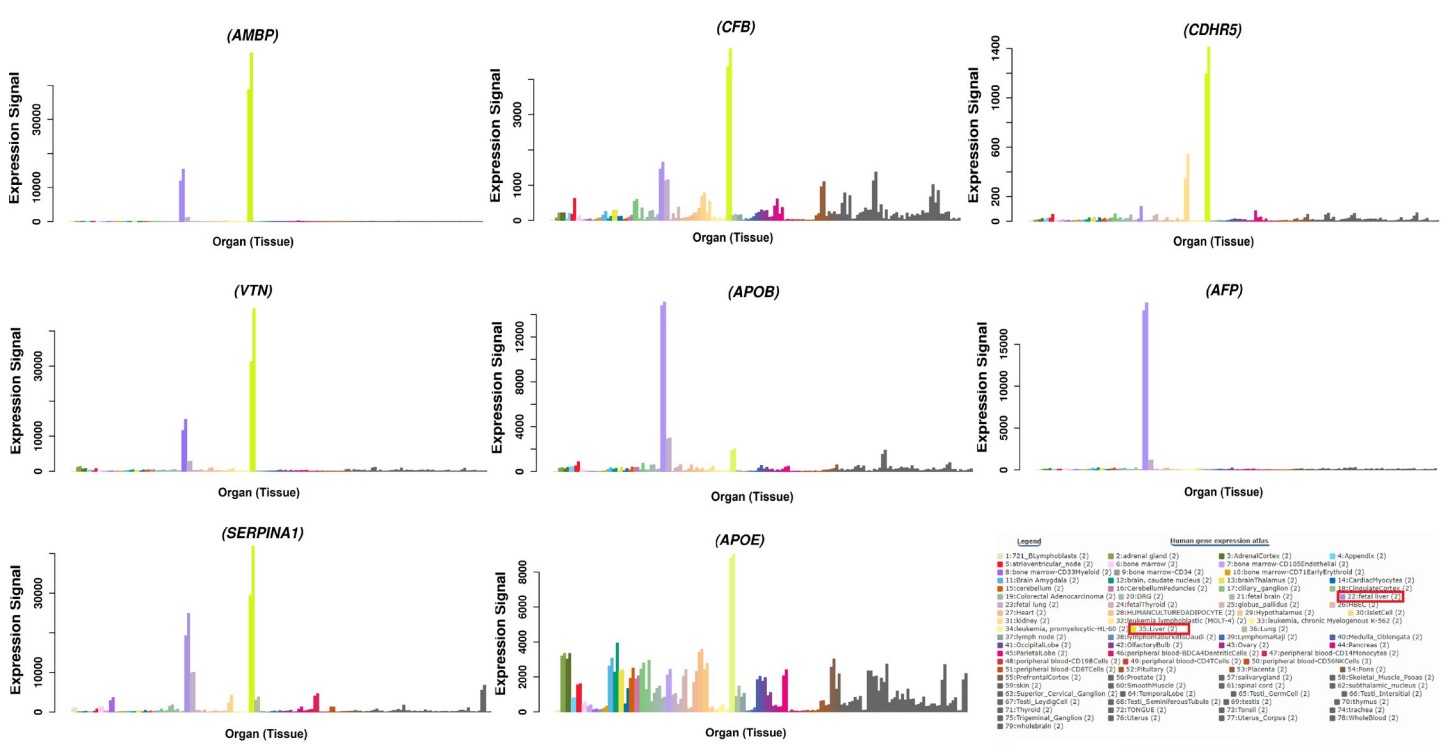

**Fig 3. Expression sites of antigenic proteins in different tissues of the human body.**

## 3.3 Prediction, screening of T-cell epitopes and conservation analysis

We predicted the multiallelic T-cell epitopes (9-mer residues) of selected proteins. Epitopes were ranked and selected on the basis of significant molecular weight, antigenicity and immunogenicity. These epitopes were antigenic having the antigenicity threshold value of >0.5 [27] and immunogenic with the significant cutoff parameter of >0.1. MHC class-I specific epitopes including LGEGATEAE, LLYIGKDRK, EDIGTEADV, QVDAAMAGR, HLEARKKSK, HLCIRHEMT, LKLSKAVHK, EQGRVRAAT (Table 2) and MHC class-II including VLGEGATEA, WVTKQLNEI, VEEDTKVNS, FTRINCQGK, WGILGREEA, LQDGEKIMS, VKFNKPFVF, VRAATVGSL were selected (Table 3).

Population coverage analysis of these multiallelic epitopes showed the effective immune response among different individuals. In conservation analysis, we found more than 90% conserved domains of these epitopes indicating the broad spectral effect (S2 and S3 Tables).

## 3.4 Safety profile analysis

We analyzed the toxicity of these epitopes based on the significant threshold value of <5 using the Support Vector Machine (SVM) platform. We observed that entire MHC class-I and II specific epitopes are non-toxic. Based on the stability score of <40, we observed that selected epitopes are stable and possessed long-term effectiveness. We analyzed that these epitopes are non-allergic and do not cause hypersensitivity and anaphylactic reactions (Tables 2 and 3).

## 3.5 Physicochemical properties of epitopes

We analyzed the hydropathicity, theoretical pI, thermostability, half-life, and solubility of epitopes. The negative value of hydropathicity (GRAVY), ionic nature and solubility score

**Table 2. Safety assessment profile of predicted class MHC class-I T-cell epitopes.**

| Sr. No. | Protein | Epitope | Antigenicity | Immunogenicity | SVM | Toxicity | Instability index | Allergenicity | Cytokine Release |
|---------|---------|---------|--------------|----------------|-----|----------|-------------------|---------------|------------------|
| 1 | AMBP | LGEGATEAE | 0.9924 | 0.2486 | -1.08 | Non-Toxic | 12.48 | Allergen | IL-6 inducer, IL-10 non-inducer, IL-4 non-inducer |
| 2 | CFB | LLYIGKDRK | 0.7433 | 0.01168 | -0.88 | Non-Toxic | -8.92 | Non-allergen | IL-6 inducer, IL-10 non-inducer, IL-4 non-inducer |
| 3 | CDHR5 | EDIGTEADV | 1.3772 | 0.25533 | -0.92 | Non-Toxic | 32.82 | Non-allergen | IL-6 inducer, IL-10 non-inducer, IL-4 non-inducer |
| 4 | VTN | QVDAAMAGR | 0.5168 | -0.02781 | -0.88 | Non-Toxic | -2.48 | Non-allergen | IL-4 inducer, IL-6 inducer, non-IL-10 inducer |
| 5 | APOBR | HLEARKKSK | 0.987 | -0.35939 | -0.97 | Non-Toxic | 8.89 | Allergen | IL-4 inducer, IL-6 inducer, non-IL-10 inducer |
| 6 | AFP | HLCIRHEMT | 0.4944 | 0.17917 | -0.33 | Non-Toxic | 27.09 | Allergen | IL-4 inducer, IL-6 inducer, non-IL-10 inducer |
| 7 | SERPINA1 | LKLSKAVHK | 0.9037 | -0.2895 | -1.43 | Non-Toxic | 16.33 | Allergen | IL-4 inducer, IL-6 inducer, non-IL-10 inducer |
| 8 | APOE | EQGRVRAAT | 1.3016 | 0.20788 | -1.14 | Non-Toxic | 30 | Non-allergen | IL-4 inducer, IL-6 inducer, non-IL-10 inducer |

indicated the hydrophilicity and water solubility of epitopes. It was observed that variations in isoelectric point substantially effects the solubility of these epitopes. Peptides with a lower or higher pI value are either positively or negatively charged at neutral pH, this charge can either increase or decrease solubility in aqueous or organic solutions. The half-life of our epitopes was ≤10 hours highlighting their significant bioavailability. We observed that these epitopes are thermostable based on the significant cutoff parameter of aliphatic index (>50) (Table 4).

### 3.6 Assessment of cytokine release

We analyzed the stimulation of effector molecules in response to check the potential effect of epitopes. Using IFNEpitope and InterleukinPred tools, we observed that these epitopes are good inducers of interferon (IFN-γ). Similarly, we predicted that these epitopes are significantly inducing interleukin-10, interleukin-6 and interleukin-4 [80] (Tables 2 and 3).

### 3.7 Proteasomal cleavage

Proteasomal cleavage pattern of antigenic proteins was investigated. Each protein possessed several cleavage sites within their sequence. Cleavage was more likely in amino acid sequences with a cleavage score higher than 0.5. The more positive peaks in graph (Fig 4) above the threshold line indicates more cleavage sites compared to the negative peaks below the line. We

**Table 3. Safety assessment profile of predicted class MHC class-II T-cell epitopes.**

| Sr. No. | Protein | Epitope | Antigenicity | SVM | Toxicity | Instability index | Allergenicity | Cytokine Release |
|---------|---------|---------|--------------|-----|----------|-------------------|---------------|------------------|
| 1 | AMBP | VLGEGATEA | 0.7458 | -1.09 | Non-Toxic | 97.78 | Non-allergen | IL-4 non-inducer, IL-6 inducer, IL10 non-inducer |
| 2 | CFB | WVTKQLNEI | 0.5072 | -1.16 | Non-Toxic | 118.89 | Allergen | IL-6 inducer, IL-10 non-inducer, IL-4 inducer |
| 3 | CDHR5 | VEEDTKVNS | 1.1645 | -0.85 | Non-Toxic | 64.44 | Non-allergen | IL-6 inducer, non-IL-4 inducer, non- IL-10 inducer |
| 4 | VTN | FTRINCQGK | 1.9257 | -0.48 | Non-Toxic | 43.33 | Allergen | IL-6 inducer, IL-4 inducer, non-IL-10 inducer |
| 5 | APOBR | WGILGREEA | 0.6057 | -0.69 | Non-Toxic | 97.78 | Non-allergen | IL-4 inducer, IL-6 inducer, IL-10 non-inducer |
| 6 | AFP | LQDGEKIMS | 0.625 | -1.16 | Non-Toxic | 86.67 | Allergen | IL-4 inducer, IL-6 inducer, IL-10 non-inducer |
| 7 | SERPINA1 | VKFNKPFVF | 0.6374 | -0.57 | Non-Toxic | 64.44 | Allergen | IL-4 inducer, IL-6 inducer, IL-10 non-inducer |
| 8 | APOE | VRAATVGSL | 0.8474 | -1.03 | Non-Toxic | 130 | Non-allergen | IL-4 non inducer, IL-6 inducer, IL-10 non inducer |

Table 4. Physicochemical parameters of Predicted class MHC class-I and II T-cell epitopes.

| | Sr. No. | Protein | Epitope | Molecular Weight (KDa) | GRAVY | Charge | Theoretical pI | Half-Life | Aliphatic Index | Solubility |
|---|---|---|---|---|---|---|---|---|---|---|
| MHC class-I | 1 | AFP | HLCIRHEMT | 1.13936 | -0.51 | -3 | 3.67 | 5.5h | 65.5 | 1.33 |
| | 2 | AMBP | LGEGATEAE | 0.875 | -0.6 | 2 | 9.7 | 5.5h | 130 | 2.18 |
| | 3 | APOBR | HLEARKKSK | 1.09642 | -0.51 | -4 | 3.43 | 1h | 86.67 | 2.82 |
| | 4 | APOE | EQGRVRAAT | 0.987 | -0.04 | 0 | 5.84 | 1h | 65.56 | 2.07 |
| | 5 | CDHR5 | EDIGTEADV | 0.947 | -2.01 | 3.5 | 10.29 | 3.5h | 54.44 | 2.08 |
| | 6 | CFB | LLYIGKDRK | 1.105 | 0.267 | 1 | 6.91 | 3.5h | 86.67 | 1.58 |
| | 7 | SERPINA1 | LKLSKAVHK | 1.02342 | -0.23 | 3.5 | 10.3 | 5.5h | 130 | 1.81 |
| | 8 | VTN | QVDAAMAGR | 0.918 | -1.03 | 1 | 9.7 | 1h | 54.44 | 1.81 |
| MHC class-II | 1 | AFP | LQDGEKIMS | 1.02017 | 0.344 | -2 | 3.79 | 10h | 97.78 | 2.02 |
| | 2 | AMBP | VLGEGATEA | 0.845 | -0.389 | 0 | 6 | 2.8h | 118.89 | 1.93 |
| | 3 | APOBR | WGILGREEA | 1.03015 | -1.222 | -2 | 4.14 | 10h | 64.44 | 1.73 |
| | 4 | APOE | VRAATVGSL | 0.873 | -0.744 | 2 | 9.51 | 1.1h | 43.33 | 1.36 |
| | 5 | CDHR5 | VEEDTKVNS | 1.020 | -0.344 | -1 | 4.53 | 2.8h | 97.78 | 2.22 |
| | 6 | CFB | WVTKQLNEI | 1.130 | -0.6 | -1 | 4.37 | 5.5h | 86.67 | 1.39 |
| | 7 | SERPINA1 | VKFNKPFVF | 1.12538 | 0.43 | 2 | 10 | 10h | 64.44 | 0.96 |
| | 8 | VTN | FTRINCQGK | 1.066 | 1.4 | 1 | 9.72 | 10h | 130 | 1.59 |

observed that these epitopes in their respective proteins were cleaved at their C-terminal site for their presentation to major histocompatibility complex molecules.

## 3.8 Development of polyvalent construct

We designed and constructed the polyvalent sequence of selected epitopes with a molecular weight of 21.43kDa containing a total number of 209 amino acid residues. The entire sequence

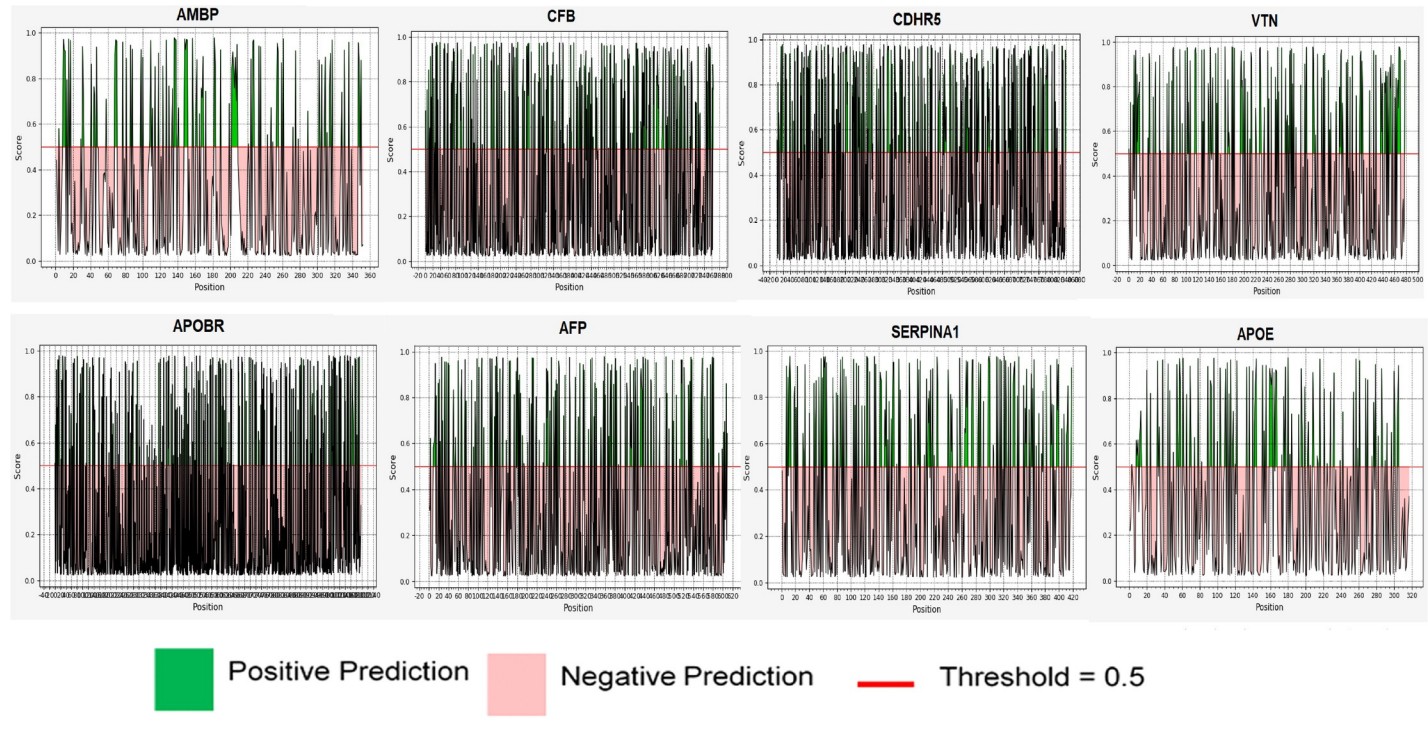

Fig 4. Proteasomal cleavage pattern of antigenic proteins.

comprises start codons, epitopes, AAY and GPGPG linkers: MAAYLGEGATEAEAAYLLYIGK
DRKAAYEDIGTEADVAAYQVDAAMAGRAAYHLEARKKSKAAYHLCIRHEMTAAYLKLSKAH
KAAYEQGRVRAATGPGPGVLGEGATEAGPGPGWVTKQLNEIGPGPGVEEDTKVNSGPGPG
FTRINCQGKGPGPGWGILGREEAGPGPGLQDGEKIMSGPGPGVKFNKPFVFGPGPGVRAATV
GSL. 3D model of this polyvalent construct was generated by I-Tesser with a confidence score
of -3.52, RMSD value of 13.8±3.9Å and Estimated TM-score of 0.33±0.11. We observed that this
polyvalent construct is thermostable and considerably antigenic (score: 0.534) and immunogenic
(score: 2.0) with a molecular weight of 21.43 kDa and pI of 7.74 similar to that of (C1) having
13.96kDa and 8.89 respectively. The instability index of the polyvalent construct based on the
significant score 21.7 indicated a stable protein complex. With an aliphatic index of 118.13, the
final anticipated vaccine construct was a highly thermostable protein. The GRAVY score of our
polyvalent construct 0.438. We analyzed the half-life of this construct was 30 hours in mamma-
lian reticulocytes, >20 hours in yeast and >10 hours in *Escherichia coli* compared to (C1) having
the same half-life of 10 hours. Based on substantial cutoff parameter (0.637 score), the solubility
analysis of polyvalent has been shown in Fig 5 indicating its water solubility at pH 7.

## 3.9 Designing of 3D models and quality analysis

3D models of antigenic proteins designed from the Robetta server were used to examine the
position of epitopes of MHC class-I (Fig 6) and II (Fig 7) highlighted as green color. We gener-
ated 3D models of each epitope by the PEP-FOLD server in PDB format to perform molecular
docking. Heatmap graph shows the structural analysis of each amino acid of epitopes. In this
graph, the probabilities of structural alphabet on vertical and horizontal axis shows helical (red
color), coil (blue) and extended configuration (green color). The QMEAN score of each epi-
tope greater than threshold value 0.5 indicating the predicted structures were similar to the
experimentally defined structures from the database (Table 5). Ramachandran plots showed
the good quality of 3D models of epitopes indicating 80% residues in favorable and other per-
missible regions (Fig 8). Similarly, 3D model of polyvalent construct was generated (Fig 9) and
we observed that the Ramachandran plot of this model showed the low-energy conformations
for psi and phi angles, as well as (59%) of total amino acid residues in favorable region (61.4%

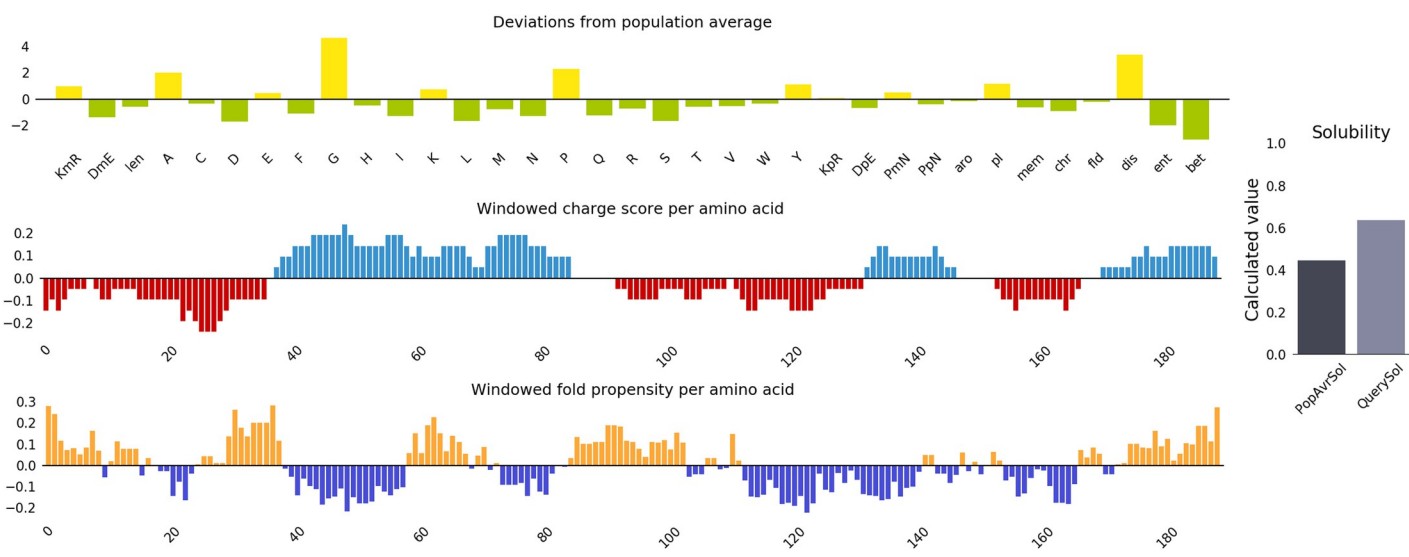

**Fig 5. Solubility analysis of polyvalent construct using protein-sol server.**

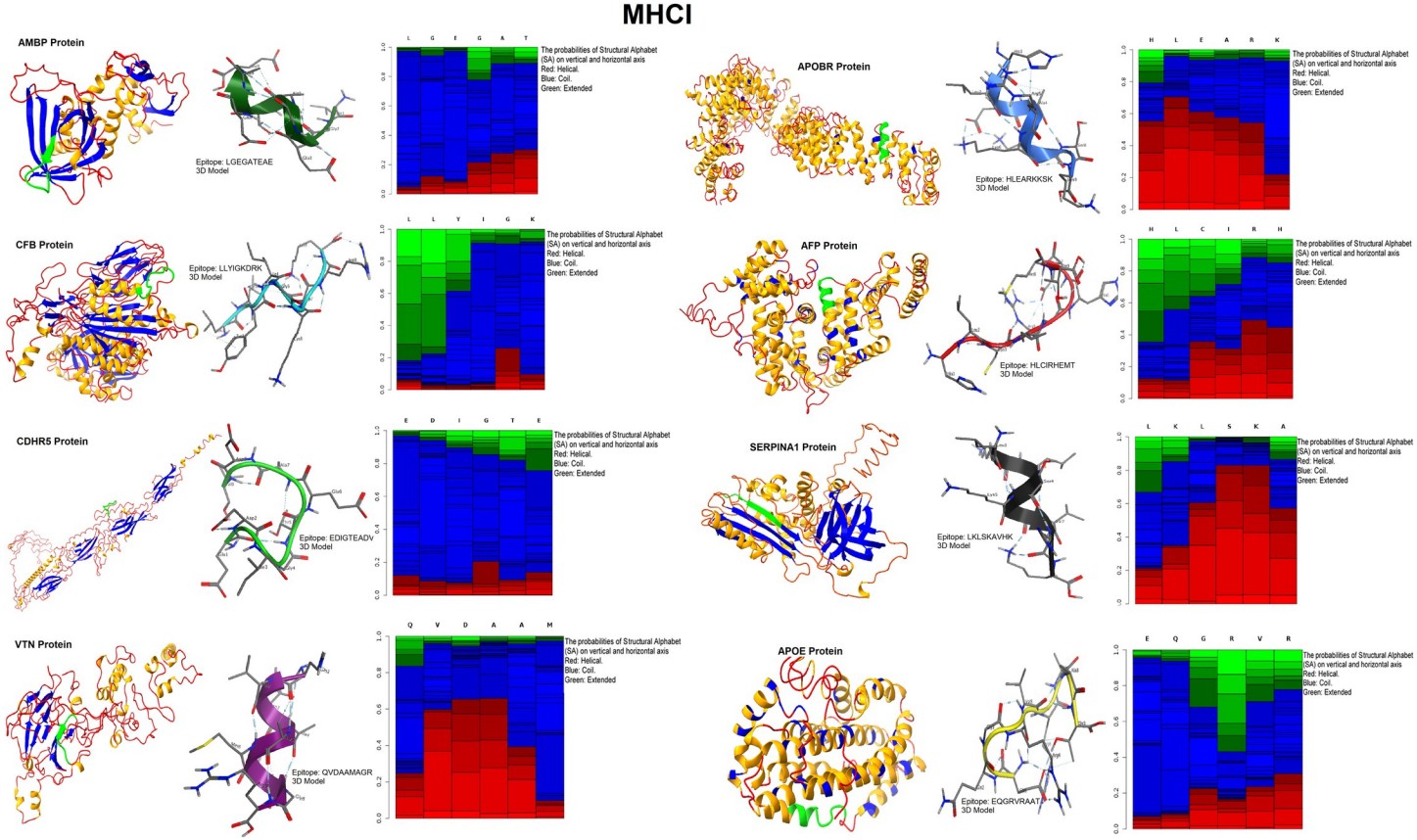

**Fig 6. 3D model of whole antigenic proteins with highlighted epitope sequence and 3D models of epitope of MHC class-I.**

in the favored region and 38.6% in the permissible region). Disulfide engineering introduce the strong covalent linkages between cysteine amino acid within the structure of protein. It helps in making the structure more rigid, stable and resistant to degradation within biological system. Disulfide engineered pairs of amino acids are shown in (Table 6) and mutated structure with yellow colored disulfide stick was given in (Fig 10).

## 3.10 Protein-Protein Interaction (PPI) network analysis

The PPI network was created in order to assess how antigenic liver cancer proteins are functionally interacting with host (human) proteins. The eight proteins AMBP, CFB, CDHR5, VTN, APOBR, FETA, A1AT and APOE have direct interaction with each other and indirect interaction with other proteins associated with liver cancer. All of these proteins interacted to form a complicated network in which nodes indicate the proteins and edges shows the interaction. In Fig 11, we segregated this network into three categories presenting antigenic liver proteins as red color nodes, yellow and purple color nodes indicate the tumor specific and tumor associated antigens, while other physiological proteins has been shown as aqua color nodes. We observed that TSAs including ALBU-HUMAN (Albumin), CTNB1-HUMAN (catenin beta-1), HEP2-HUMAN (Heparin cofactor-2), ANT3-HUMAN (antithrombin-III), A2MG-HUMAN (alpha-2 macroglobulin), USH1C-HUMAN (harmonic) and TAAs comprising FETUA-HUMAN (alpha-2-HS glycoprotein), FIBB-HUMAN (fibrinogen beta chain), FITM2-HUMAN (acyl-coenzyme A diphosphatase), PRDX4-HUMAN (peroxiredoxin-4) have direct functional relationship with target proteins.

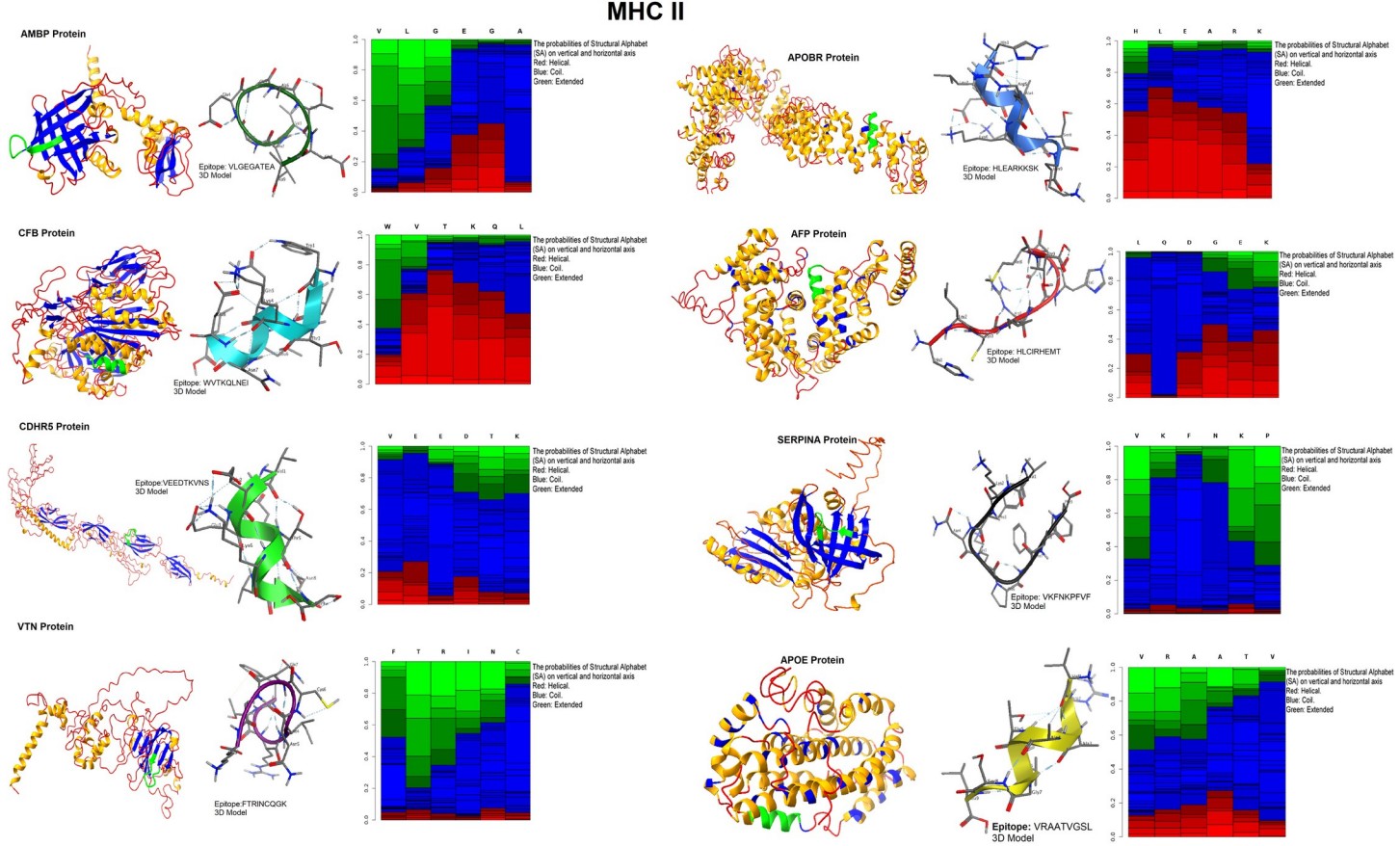

**Fig 7. 3D model of whole antigenic proteins with highlighted epitope sequence and 3D models of epitope of MHC class-II.**

## 3.11 Molecular docking of epitopes and polyvalent construct

The *in-silico* binding affinity (E-score) of the epitopes with MHC molecules (class I and II) was analyzed using MOE software. The binding affinities of the MHC class-I epitopes including LGEGATEAE, LLYIGKDRK, EDIGTEADV, QVDAAMAGR, HLEARKKSK, HLCIRHEMT, LKLSKAVHK, EQGRVRAAT were observed as: -9.57, -8.86, -8.10, -7.99, -8.73, -9.01, -8.51, -8.56 kcal/mol respectively. We observed these epitopes showed interaction with amino acid residues: arginine-A55, glycine-B55, glutamine-A154, asparagine-B53, glutamic acid-A63, and tyrosine-A159 of the MHC-I receptor (Fig 12). Similarly, binding energies of MHC class-II

**Table 5. QMEAN score of predicted class MHC class-I and II T-cell epitopes.**

| Sr. No. | Proteins | MHC-I epitope | QMEAN score | MHC-II epitope | QMEAN score |
|---|---|---|---|---|---|
| 1 | AMBP | LGEGATEAE | 0.558 | VLGEGATEA | 0.526 |
| 2 | CFB | LLYIGKDRK | 0.542 | WVTKQLNEI | 0.937 |
| 3 | CDHR5 | EDIGTEADV | 0.497 | VEEDTKVNS | 0.591 |
| 4 | VTN | QVDAAMAGR | 0.688 | FTRINCQGK | 0.493 |
| 5 | APOBR | HLEARKKSK | 0.848 | WGILGREEA | 0.569 |
| 6 | AFP | HLCIRHEMT | 0.584 | LQDGEKIMS | 0.644 |
| 7 | SERPINA1 | LKLSKAVHK | 0.679 | VKFNKPFVF | 0.645 |
| 8 | APOE | EQGRVRAAT | 0.422 | VRAATVGSL | 0.458 |

**MHC I**

**MHC II**

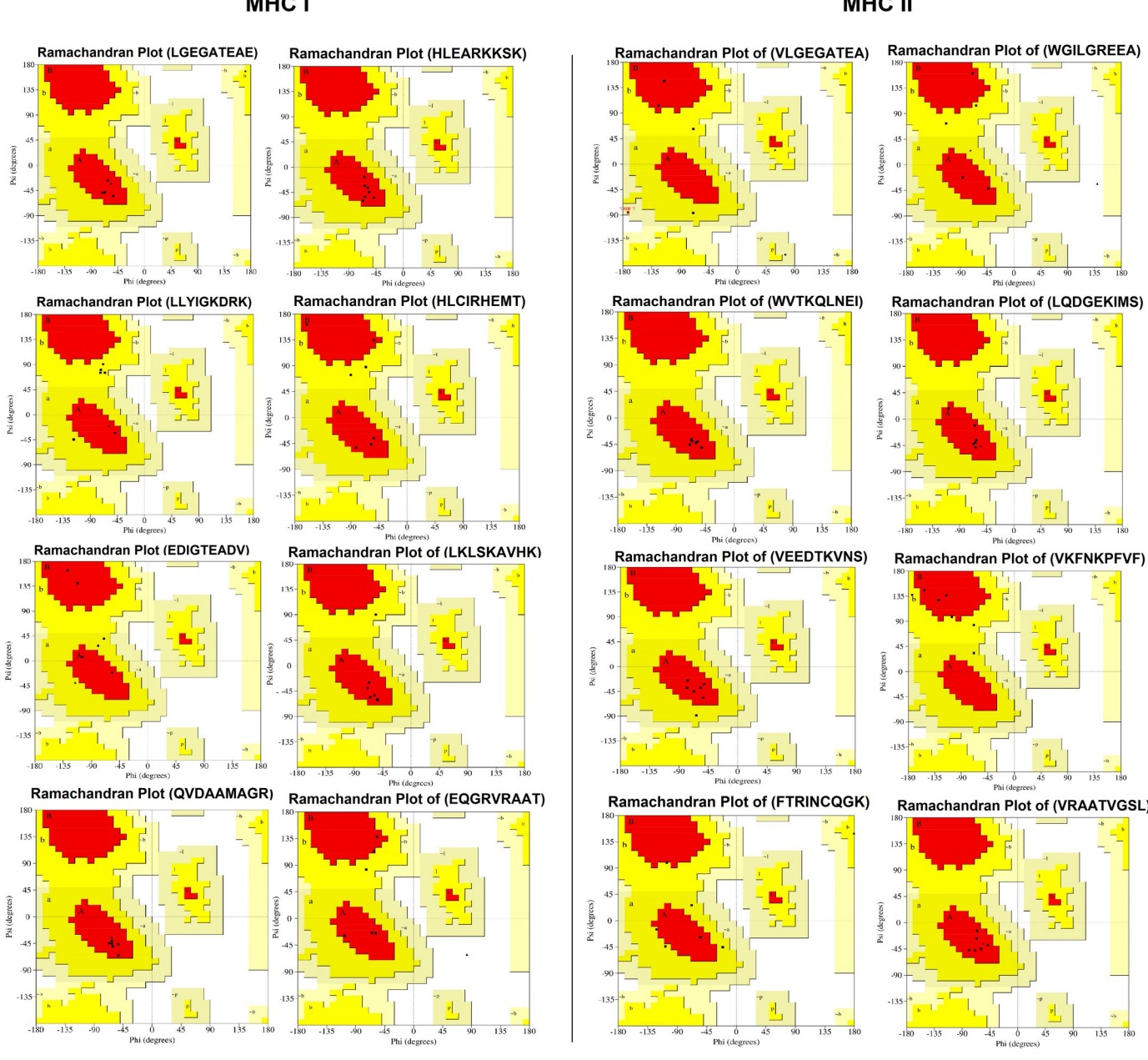

**Fig 8. Ramachandran plots representing the quality of predicted MHC class-I and II T-cell epitopes.**

epitopes including VLGEGATEA, WVTKQLNEI, VEEDTKVNS, FTRINCQGK, WGILGREEA, LQDGEKIMS, VKFNKPFVF and VRAATVGSL are -9.20, -11.92, -10.97, -11.16, -10.20, -10.24, -10.06 and -9.82 kcal/mol respectively. In interaction, we found glutamine-A28, glutamic acid-B35, asparagine-C83, leucine-C144, threonine-C80, valine-B50, lysine-A111, arginine-A140, tyrosine-B32, proline-C85, and serine-B54 amino acid residues (Fig 13).

We performed molecular docking of polyvalent construct using ClusPRo software binding energy (E-score) of polyvalent with MHC class-I and II was -1072.3 and -1107.3 respectively. Ligand interactions of construct with MHC class-I and II developed by LigPlot tool. Aspatic acid-122, glutamine-A96-32, arginine-A32-3-234, tyrosine-A27, histidine-A192, threonine-

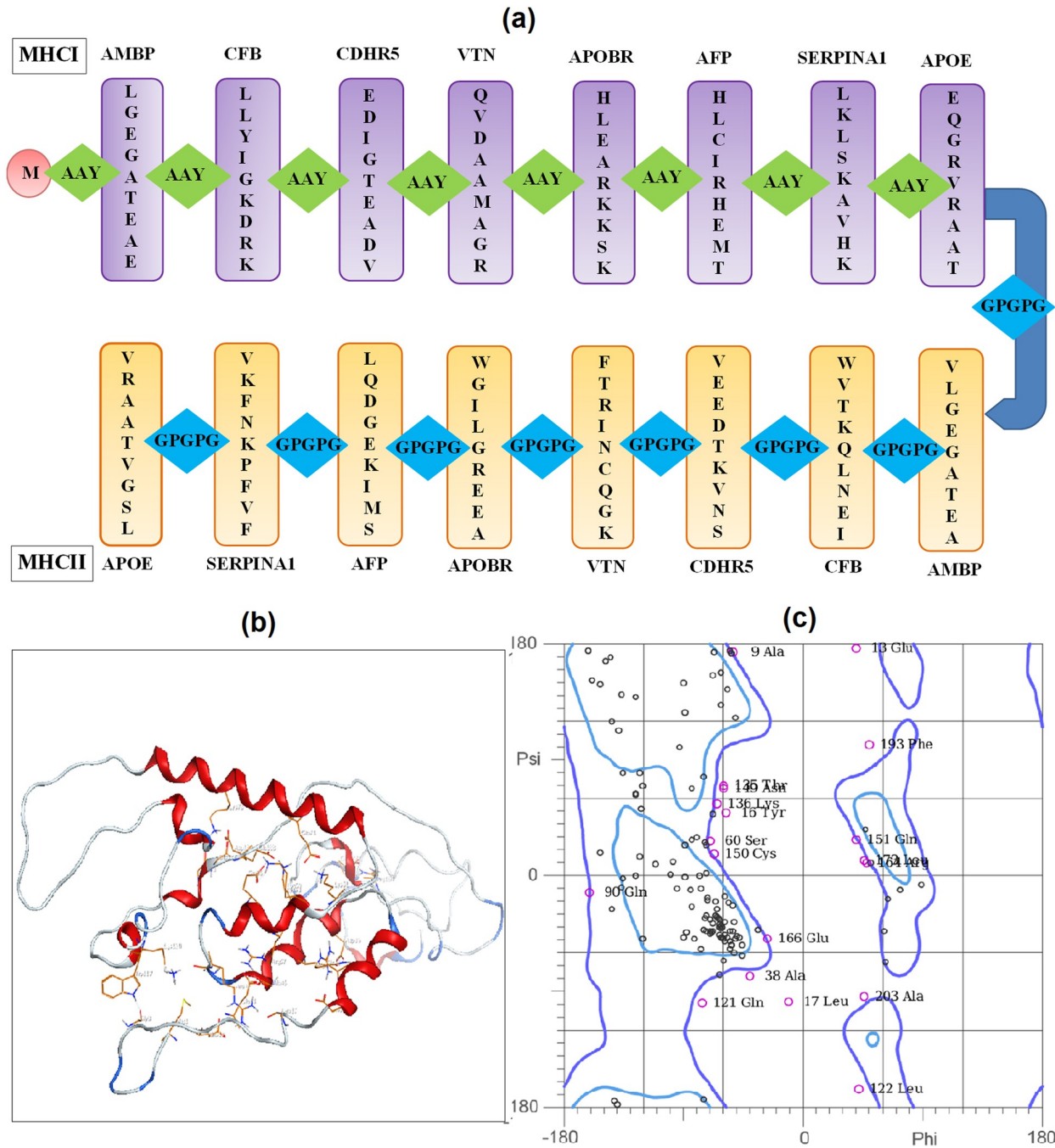

**Fig 9.** (a) Schematic representation of polyvalent construct having 209 residues (b) 3D model of construct obtained from I-Tesser (c) Ramchandran plot indicates the quality of construct.

A190, arginine-A202, glutamine-A232-242, tryptophan-A232, proline-A235, alanine-A236, aspartic acid-A238 are the residues found in the interaction. Green and red dashed lines represent hydrogen bonds and salt bridges respectively. The residues involved in hydrophobic interactions are shown with a half circle in red color (Fig 14).

**Table 6. Modified pairs of amino acid residues, torsion (χ₃) angles and energy values of disulfide engineering.**

| Amino Acid Residues Pairs | $\chi_3$ | Energy(kcal/mol) |
|---|---|---|
| LYS61-LEU66 | -109.57 | 4.00 |
| LEU77-SER80 | +111.88 | 1.81 |
| ALA50-LEU77 | +110.11 | 3.89 |
| TYR76-ALA95 | -103.46 | 2.45 |

### 3.12 Molecular dynamics simulation of polyvalent construct

We used the Groningen Machine for Chemical Simulations (GROMACS) spectro built-in tool to check the stability of the polyvalent vaccine construct complexed with MHC class-I and II. During MD simulation, various parameters such as pressure, temperature, radius of gyration (Rg), root-mean-square fluctuation (RMSF) and root-mean-square density (RMSD) were measured. Optimized Potential for Liquid Simulation (OPLS) force-field was used to control the protein in cubic box for solvation of the protein in water. Genion, a tool of GROMACS applied to neutralize the total charge of the protein using a cutoff scheme (Verlet) and for this purpose 22 Na⁺ ions and 14 Cl⁻ were added in protein complex of MHC class-I and II respectively. R-programme was used to generate graphs. In case of MHC class-I, the average potential energy of the system was measured -1.58149e$^{+5}$, at temperature 300.22˚K, and the total pressure of 2.042 bar and for MHC class-II, average potential energy was -1.58149e$^{+5}$, temperature and average pressure was 299.94˚K and 5.68 bar respectively. After the equilibration of system, we found the total average density of 1019.05 kg/m³ and 1009.327846 for MHC class-I and II (Fig 15). A radius of gyration for RMSD and RMSF was determined by generating analysis of trajectory after a 50-ns simulation. Rg value of the graph (19.8 to 28.8) indicated that the residues of receptor-ligand complex are closer to each other to make a compact configuration for stable interactions. RMSD graph of simulation reaches a plateau condition indicating the simulation time is sufficient for this protein and confirms the equilibrium of the system.

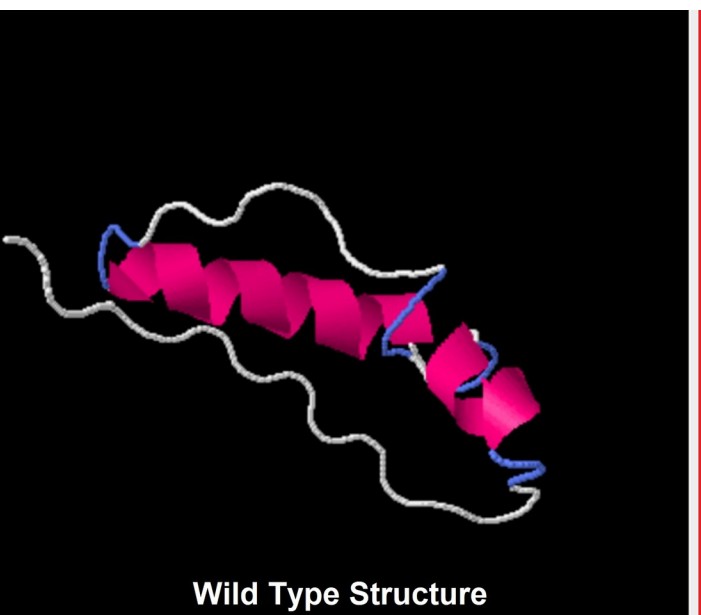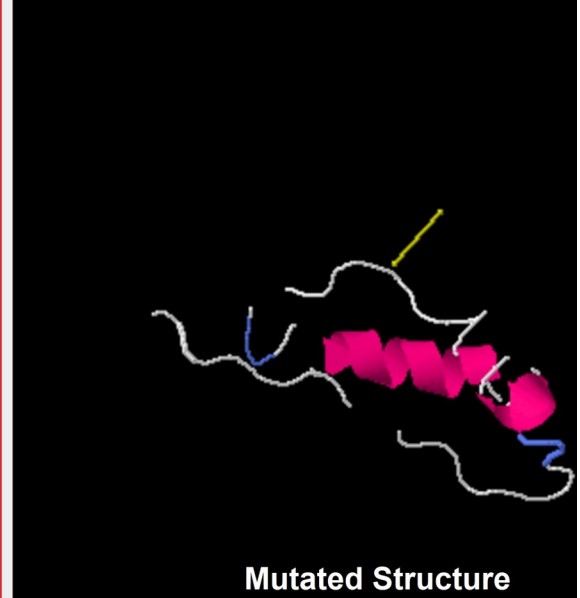

**Fig 10. Disulfide engineering of polyvalent construct indicating original model and mutated model.**

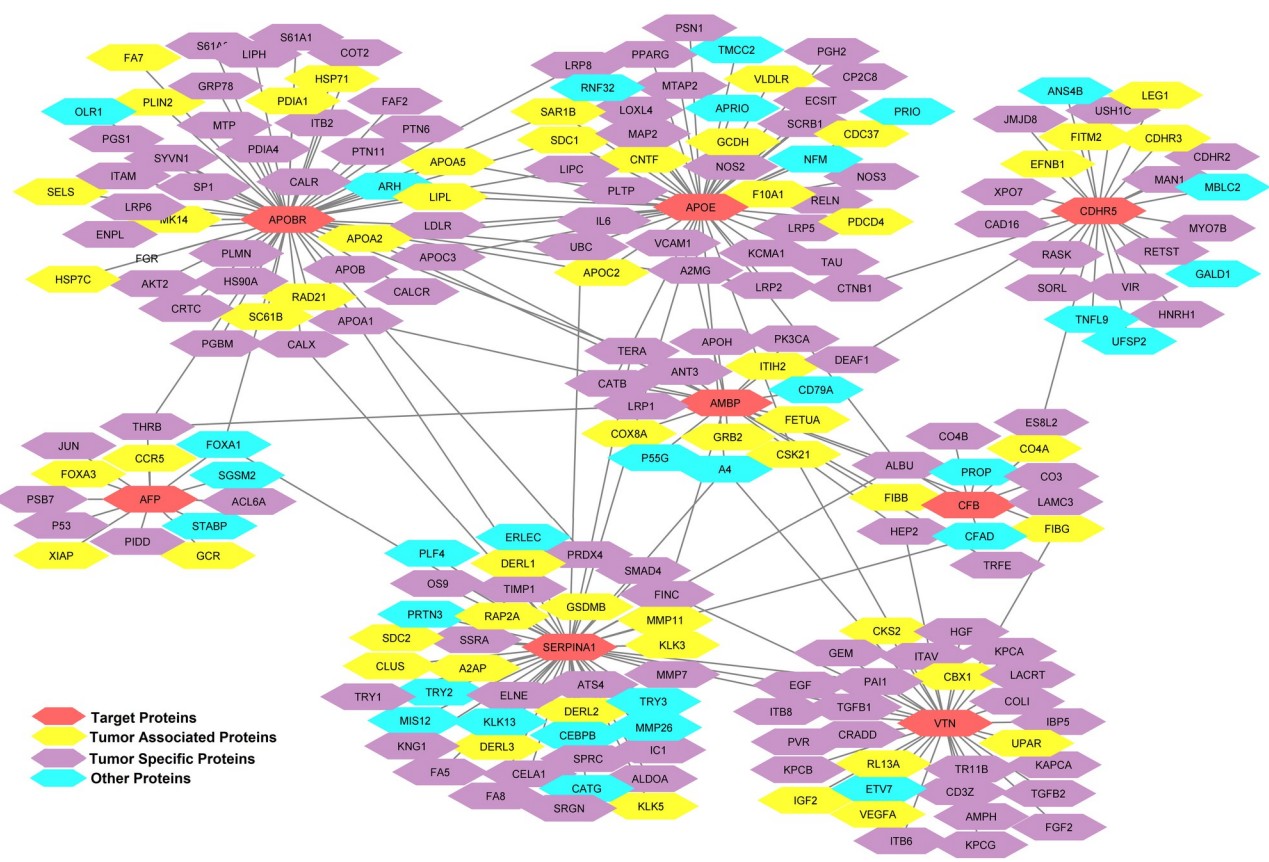

**Fig 11. Protein-protein interaction network of our target proteins with other host proteins.**

During simulation, we observed that RMSD plot showed the structure stability presenting the sustainable score at 58Å. Additionally, we analyzed no variations in RMSD highlighting the stable complex. We used RMSF to assess the fluctuation in amino acid residues of protein complex and observed that target proteins are responsible for turns and coils in the entire structure. It is demonstrated that the protein complex of our polyvalent construct in its final form has good stability in the solvated dynamic state.

### 3.13 Immune simulation

During immune simulation, we found significant immunoglobulin titers after primary and secondary injections of the vaccine construct. In Fig 16A, initially, we observed a substantial increase of IgM antibody concentration. Antibody levels (IgM, IgG, IgG1, and IgG2) increases as a result of secondary and tertiary responses during the simulation period and antigen level is reduced. The population of B cells, including memory B cells, increased significantly (Fig 16B) indicated the possibility of memory formation and isotype switching while we observed the increase in B cell proliferation and antigen presentation after immunization (Fig 16C). In response to memory development, TH (helper) and TC (cytotoxic) cell populations was significantly high as shown in Fig 16D–16F. During the secondary vaccine injection, we found higher titer rate of macrophage activity and antigen presentation cells (Fig 16G). After the third injection of vaccine, we analyzed a significant increase of interferon gamma (IFN-γ) and a moderate increase in interleukin 2 (IL-2) (Fig 16H). These findings showed the potential impact of polyvalent construct.

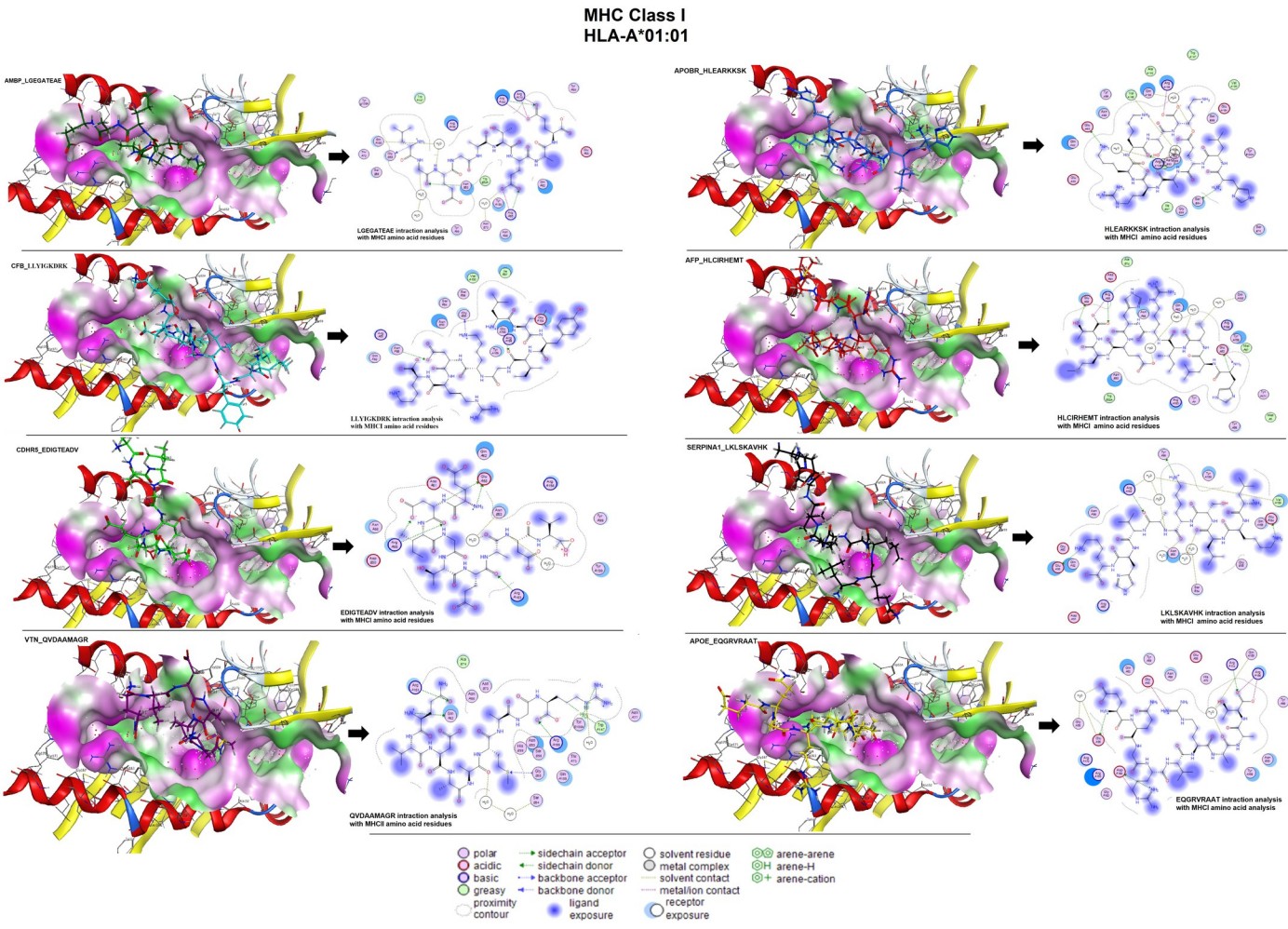

**Fig 12. Molecular docking and ligand interaction analysis of individual epitopes of MHC class-I with their target molecules.**

### 3.14 Codon optimization and in silico cloning

*In silico* cloning was used to validate the expression of polyvalent construct in *E. coli* strain. This construct was optimized using JCAT server containing 639 nucleotides with a GC content of 55.5% (30%-70%), and a codon adaptation index (CAI) value of 0.965 (0.8–1.0). Restrictions sites BamHI and HindIII were inserted at the C and N terminals of the optimized polyvalent construct. This modified construct was then cloned at MCS (multiple cloning sites) between these specific restriction sites of the pET30a (+) vector. The resulting clone has a total length of 5977bp (Fig 17).

## 4. Discussion

Liver cancer is the fourth most lethal disease responsible for 8.2% of deaths globally. It is a major health and economic burden. Traditional treatment methods like surgery, drugs, and combined therapies have some limitations [81]. This research proposes more economical and resourceful integrated immunoinformatic and reverse vaccinology approaches to find potential liver cancer vaccine candidates. In this regard, T-cell epitopes predictions are more effective towards vaccine development against liver cancer [82].

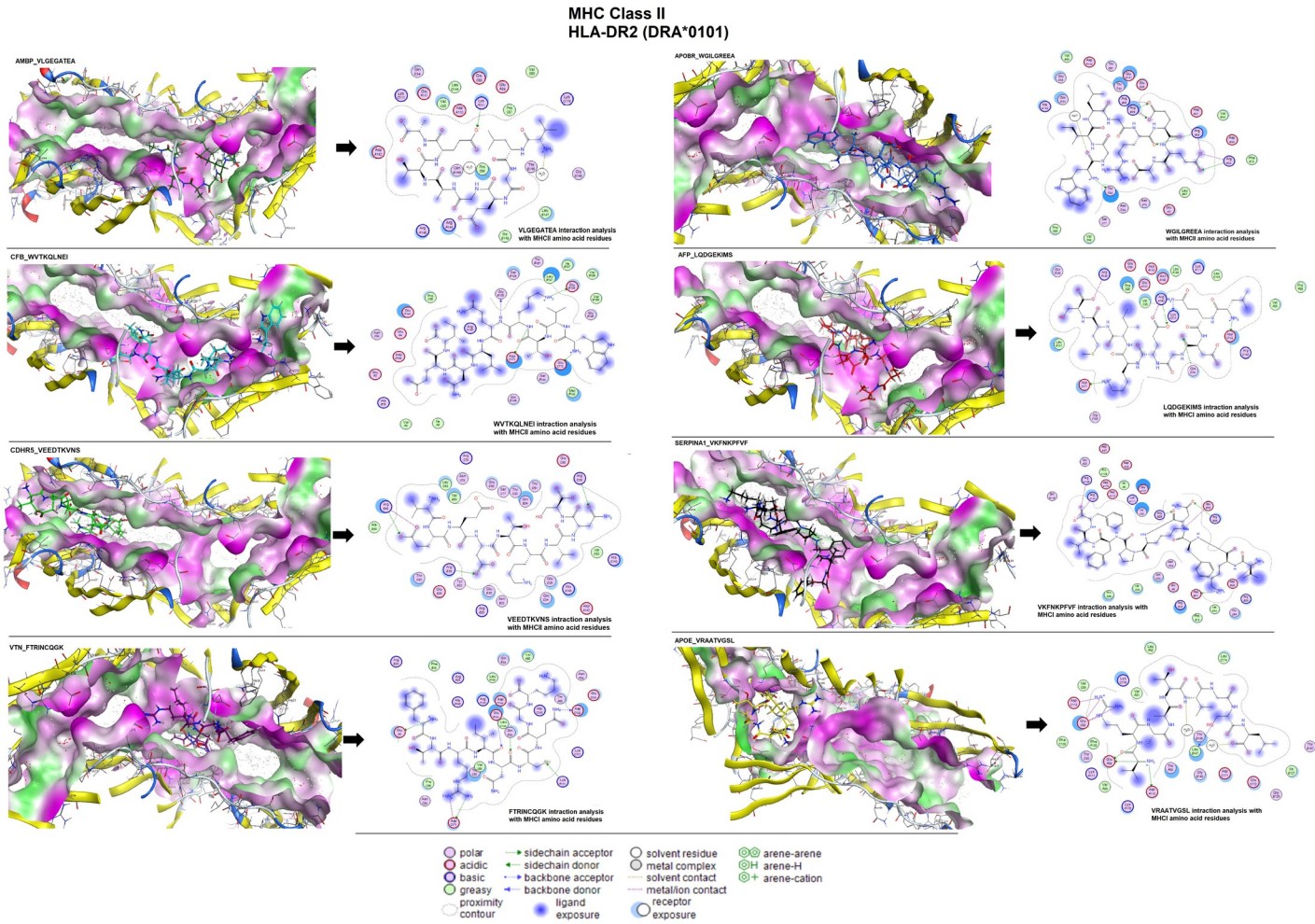

**Fig 13. Molecular docking and ligand interaction analysis of individual epitopes of MHC class-II with their target molecules.**

In our study, we used a comprehensive immunoinformatic based framework to design a therapeutic vaccine against the liver cancer. We started screening from 125 liver cancer proteins and finally found 8 proteins which are antigenic, extracellular and highly expressed in liver tissues, making them easily accessible to the immune system [83].

We focused on the prediction of 8 MHC class-I and II T-cell epitopes. MHC class-I epitopes were 100% conserved across various populations of the selected proteins, with the exception of HLEARKKSK, which had a probability of SNP occurring at location H135Y, making them perfect targets for vaccine development [84]. Three epitopes of MHC class-II (WGILGREEA, VLGEGATEA, VEEDTKVNS) have some SNP variations with probability of more than 20%. The study explained by Sukumar *et al* [85] claimed, these variations are still relatively infrequent and may not significantly impact the effectiveness of epitopes. Overall, the high conservation of these epitopes suggests they are promising candidates for a broadly effective liver cancer vaccine.

Pedro Romero *et al* [86] and I Farhani *et al* [87] implemented some bioinformatic tools to predict the antigenicity and immunogenicity of peptides against human melanoma and found three peptides give the best results on the basis of cutoff values of antigenicity and immunogenicity, similarly our predicted T-cell epitopes show antigenicity scores exceeding 0.5 and

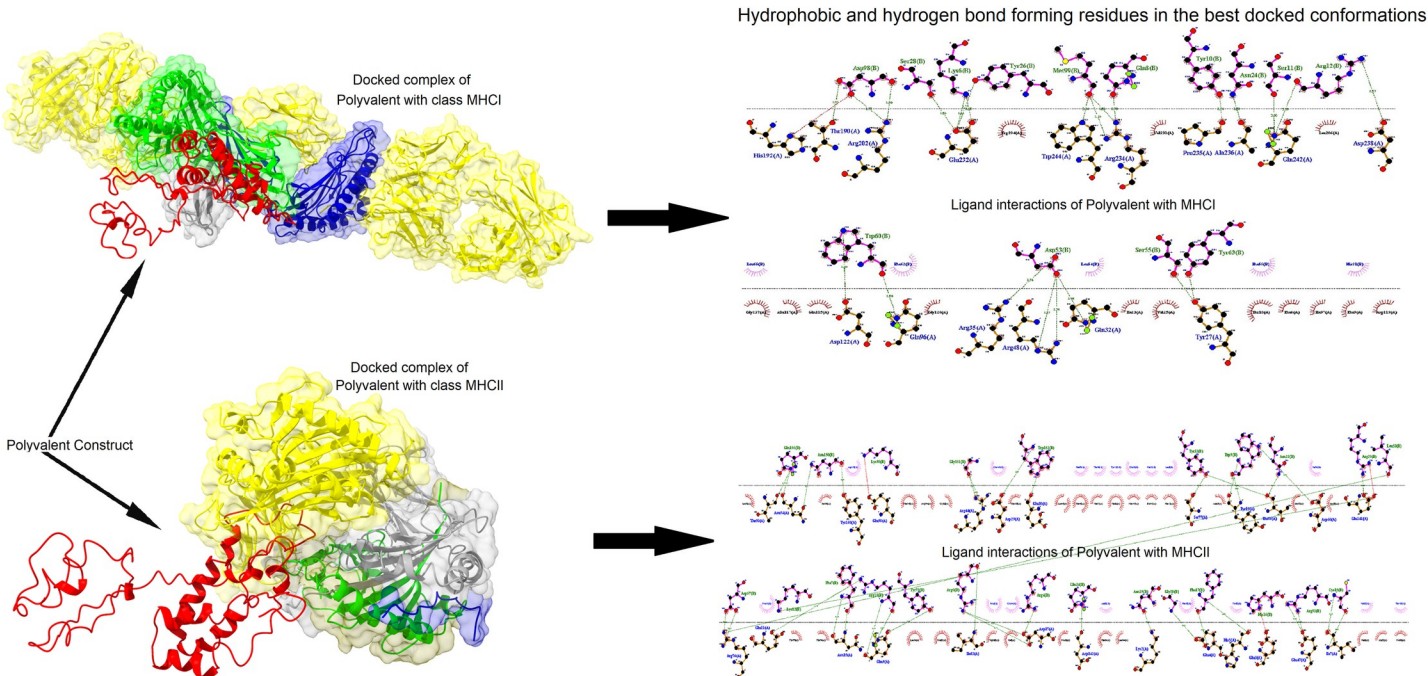

**Fig 14. Docking analysis of polyvalent construct with MHC class-I and II target molecules along with hydrophobic ligand interaction.**

immunogenicity >0.1 indicating their potential to generate a strong immune response [88]. VaxiJen is a first protein antigenicity prediction tool for bacteria, viruses and tumor. In this server, a significant cutoff threshold of 0.5 is set for all predicted epitopes to get the best possible outcomes amongst different antigens [27]. In a study conducted by Maolin *et al* [89], authors found 70 unique peptides based on toxicity profile and allergenicity as an important screening parameter. According to AllerTop v. 2.0's allergenicity prediction [90], we selected non-allergenic epitopes to reduce the possible risk of allergic reactions. Additionally, physico-chemical characteristics and safety profile studied by Chiangjong *et al* [91] and Rathore *et al* [92] reported stable, safer and effective cancer peptides and highlighted the significance of the physicochemical properties analysis and toxicity prediction. We observed the parallel outcomes indicating the substantial physicochemical properties including non-toxicity, half-life, water solubility, and stability [93]. Similarly, these peptides are capable to activate effector molecules including cytokines and antibodies [91, 94].

For simulation, molecular modeling is important and for this purpose, we used ERRAT server to examine the quality of 3D models of epitopes [95]. Each epitope demonstrated a significant proportion of residues (40% to 100%) in favored and allowed regions indicating overall good geometry [96]. Remarkably, none of the residues were observed in disallowed regions, implying that the predicted structures are unlikely to contain major structural errors. Likewise, all epitopes had positive Q-mean scores, indication a strong level of similarity between the predicted and experimentally determined protein structures [97]. These findings provided further evidence to support the validity and potential effectiveness of these epitopes as vaccine candidates.

According to Vilar *et al* [98], molecular docking is effective to screen the vaccine candidates and we observed that the interaction of epitopes and HLA (MHCs) molecules based on their binding energies showed significant binding affinities. The computed binding energies of the MHC-I specific epitopes range from -7.93 to -10.09 kcal/mol and MHC-II specific epitopes

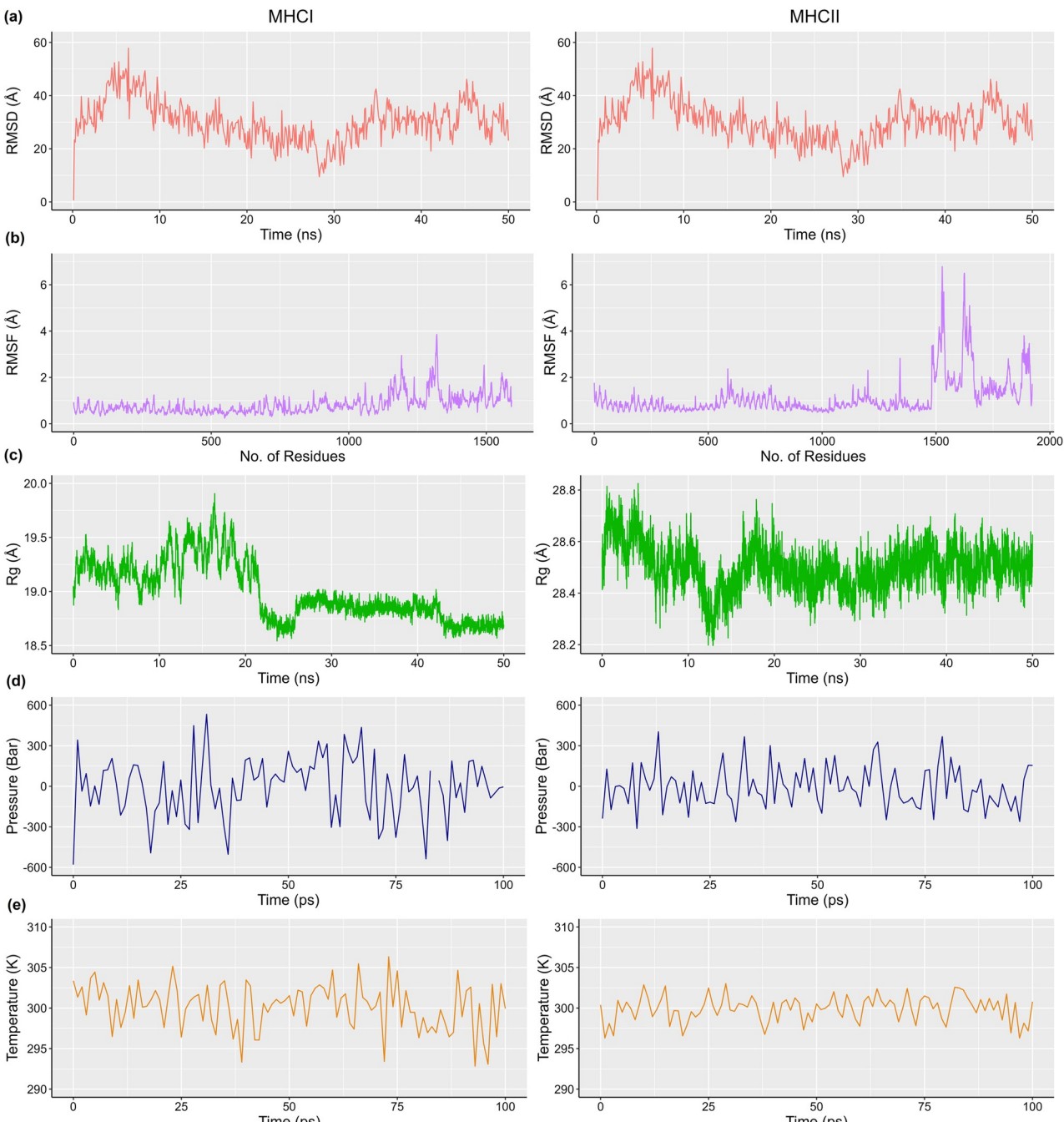

**Fig 15.** MD simulation of polyvalent vaccine construct with MHC class-I and II (a) Temperature of the system maintained around 300°K (b) Pressure plot of ligand at equilibrium at 100ps (c) Rate of gyration indicating the stability of construct throughout the simulation time (d) RMSD level rises up to 57.87Å indicating the construct's stability (e) RMSF plot's peaks indicating the flexibility of complex.

showed -9.26 to -11.15 kcal/mol presenting considerable interactions between these epitopes and target molecules.

We analyzed that potential polyvalent vaccine candidates could be effective against cancer. In previous studies, it has been reported that such polyvalent candidates showed substantial activity against Wilms tumor-1 (WT1) and human papillomavirus (HPV) cancer [17] against

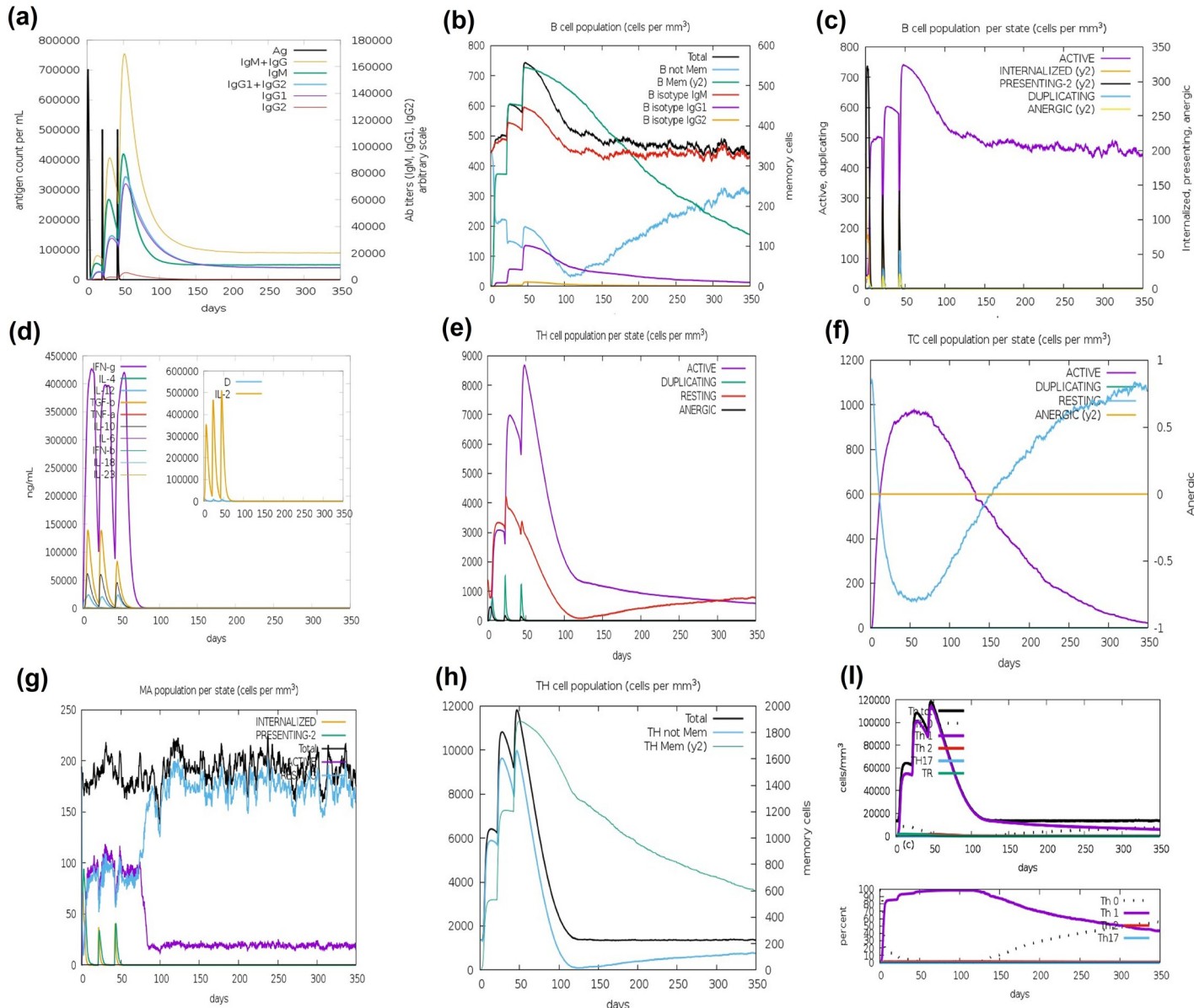

**Fig 16.** Immune simulation by C-ImmSimm (a) Different types of immunoglobulins are represented by different colors as response to vaccine injection (b) Post injection population of B cell population (c) Per state of B-cell population (d) Helper T-cell population (rise in memory cells) (e) Per state population of T-helper cell (f) Post injection increase in cytotoxic T-cells (g) Per state population of macrophages (h)Significant rise in titer of IFN- and IL-2Production (i) Th1 production.

Hepatitis C [99]. Similarly, polyvalent vaccine was designed to analyze the broad spectral effect against respiratory syncytial virus (RSV) using simulation based framework [100]. In our analysis, our polyvalent construct containing 209 amino acid residues including AAY and GPGPG linkers with a molecular weight of 21.43kDa is significant immunogenic 17, [38, 101]. We observed the considerable binding energies (-1072.3 and -1107.3) during molecular interaction. MD simulations were performed to evaluate the stability of the docked complexes with MHC class-Iand II molecules using different statistical algorithms as previously described [102, 103]. *In silico* cloning is important to develop cost effective vaccines [104]. The codon adaptation index (CAI) value indicated [105] that the polyvalent mRNA codons were

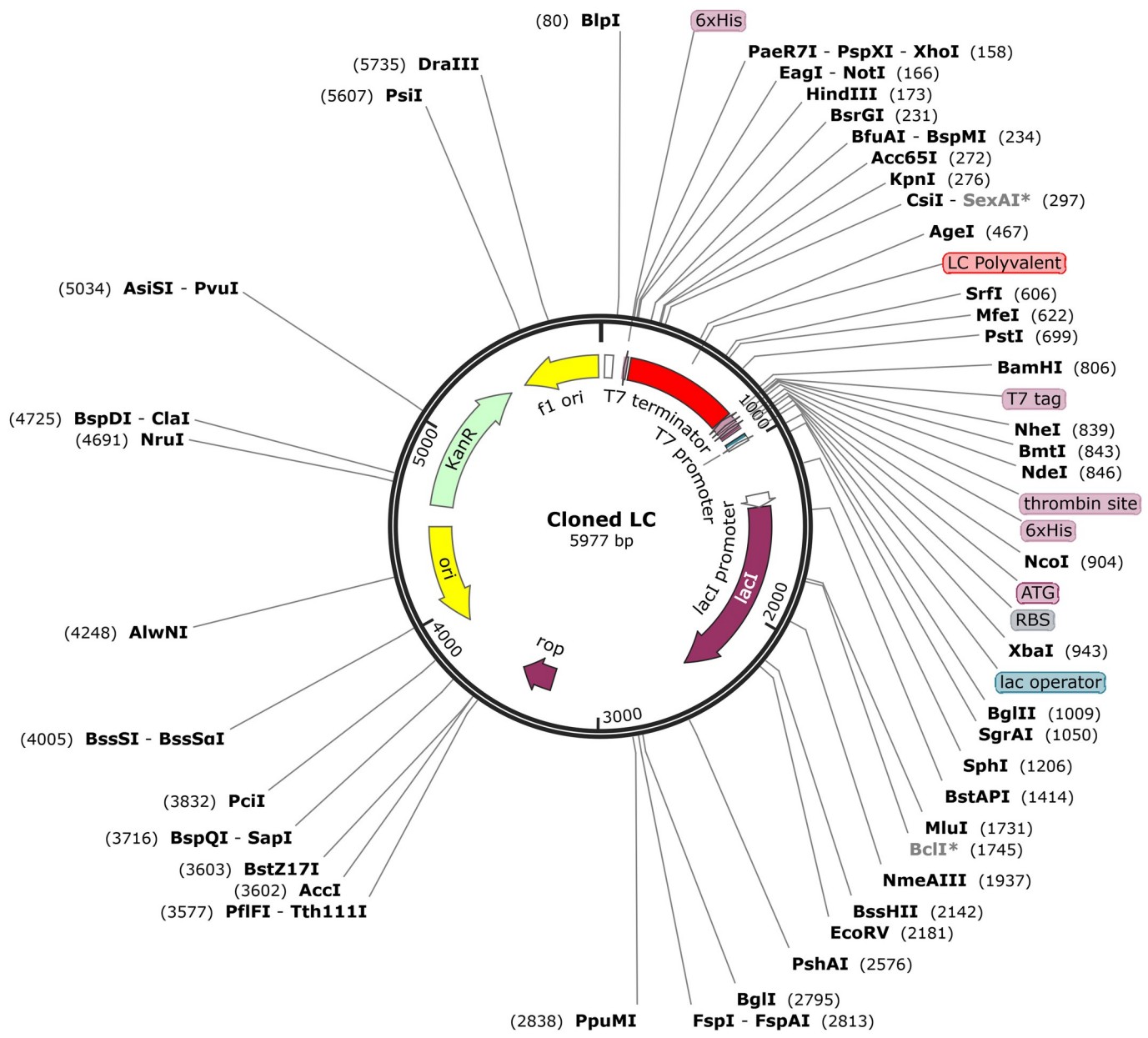

**Fig 17. *In silico* cloning of polyvalent vaccine construct into pET28a (+) plasmid between restriction sites BamHI and HindIII.**

compatible with the *E. coli system* suggesting that the predicted polyvalent production was possible by *in silico* cloning [106]. The selected epitopes and potential polyvalent construct are efficiently recognized and presented by the immune system triggering an immune response against liver cancer.

## 5. Conclusion

Immunotherapy against cancer stimulates the patient's immune system to recognize and eliminate cancerous cells. It is crucial to determine the antigenic proteins present on malignant cells. The application of this computational approach led to the identification of potent MHC

class-I and II T-cell epitopes that potentially be used in the development of a vaccine against liver cancer. This research study suggests that these potential epitopes have significant binding affinity and immunogenicity to produce a therapeutic immune response against liver cancer.

## Supporting information

**S1 Table. List of bioinformatics tools/ software used in this research study.**
(DOCX)

**S2 Table. Prediction, conservational analysis and population coverage of multiallelic MHC class-I T-cell epitopes.**
(DOCX)

**S3 Table. Prediction, conservational analysis and population coverage of multiallelic MHC class-II T-cell epitopes.**
(DOCX)

**S1 Graphical abstract.**
(TIF)

## Author Contributions

**Conceptualization:** Syed Aun Muhammad.

**Data curation:** Syed Aun Muhammad, Haris Khurram, Iraj Muqaddas, Rehan Sadiq Shaikh, Baogang Bai.

**Formal analysis:** Yuhe Bai, Syed Aun Muhammad, Jinlei Guo, Baogang Bai.

**Investigation:** Jinlei Guo, Saba Zafar, Iraj Muqaddas, Rehan Sadiq Shaikh, Baogang Bai.

**Methodology:** Sidra Zafar, Syed Aun Muhammad.

**Software:** Yuhe Bai, Jinlei Guo, Haris Khurram, Baogang Bai.

**Supervision:** Syed Aun Muhammad.

**Validation:** Baogang Bai.

**Visualization:** Sidra Zafar, Syed Aun Muhammad, Saba Zafar.

**Writing – original draft:** Sidra Zafar, Syed Aun Muhammad, Iraj Muqaddas.

**Writing – review & editing:** Syed Aun Muhammad.

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
