## [Decision Letter · Decision Letter 0]

25 Jun 2024

PONE-D-24-20487Molecular Dynamics Simulation based Prediction of T-cell Epitopes for the Production of Effector Molecules for Liver cancer immunotherapyPLOS ONE

Dear Dr. Muhammad,

Thank you for submitting your manuscript to PLOS ONE. After careful consideration, we feel that it has merit but does not fully meet PLOS ONE’s publication criteria as it currently stands. Therefore, we invite you to submit a revised version of the manuscript that addresses the points raised during the review process.

We look forward to receiving your revised manuscript.

Kind regards,

Sheikh Arslan Sehgal, PhD

Academic Editor

PLOS ONE

Journal Requirements:

Reviewers' comments:

Reviewer's Responses to Questions

**Comments to the Author**

1. Is the manuscript technically sound, and do the data support the conclusions?

Reviewer #1: Yes

Reviewer #2: Partly

2. Has the statistical analysis been performed appropriately and rigorously? 

Reviewer #1: N/A

Reviewer #2: N/A

3. Have the authors made all data underlying the findings in their manuscript fully available?

Reviewer #1: Yes

Reviewer #2: Yes

4. Is the manuscript presented in an intelligible fashion and written in standard English?

Reviewer #1: Yes

Reviewer #2: No

5. Review Comments to the Author

**Reviewer #1:** 1.In the materials and methods section, the authors should put the access link in front of the names of the software, servers, and databases.

2.The authors should explain why they set the antigenicity threshold to 0.5 in the manuscript. Do they have a reference for this value?

3.Please calculate and report the aliphatic index and GRAVY parameters of the vaccine construct.

4.The authors should perform disulfide engineering of the vaccine construct.

5.Please compare the characteristics of the vaccines designed in this study with the vaccines designed in other authors' studies. For this, please use the following references and cite them.

https://doi.org/10.3390/vaccines11020263

https://doi.org/10.2174/1573409919666230612125440

https://doi.org/10.1080/07391102.2023.2258403

https://doi.org/10.1016/j.ijbiomac.2024.131517

https://doi.org/10.1007/s12033-023-00949-y

**Reviewer #2: **The manuscript is a bioinformatic analysis of TSA and TAA for Liver cancer. Although the immunoinformatic analysis, molecular dynamics analysis, immune simulation studies are also sufficient to predict the usage of this epitope based vaccine but unless it is validated by using cancer cell lines or patients cells, the results cannot be confirmed. Animal model studies with immunogenic response and tumour regression is required to further validate their claim. So many such vaccine candidates are predicted for almost all the diseases and cancers speacially in COVID times and are stll being predicted, but they need to be validater preclinically or experimentally.

6. PLOS authors have the option to publish the peer review history of their article (what does this mean?). If published, this will include your full peer review and any attached files.

Reviewer #1: No

Reviewer #2: **Yes: **Sadhna Sharma

---

## [Author Response · Author response to Decision Letter 0]

27 Jul 2024

July 22, 2024

The Editor,

PLOS ONE 

Subject: Re-Submission of the revised manuscript.

Manuscript No. ID. PONE-D-24-20487

Title: “Molecular Dynamics Simulation based Prediction of T-cell Epitopes for the Production of Effector Molecules for Liver cancer immunotherapy”

We are thankful to the editorial board for useful comments and suggestions which helped us in improving our manuscript. Further suggestions would be appreciated if any.

We have modified the manuscript accordingly and individual responses to reviewer’s comments are listed below point by point:

Report

Reviewer 1

(1) In the materials and methods section, the authors should put the access link in front of the names of the software, servers, and databases.

Author’s Response

We are thankful for reviewer comments. As per recommendations of reviewer’s, Access links of all the software used in this research are added into the Supplementary Table 1 along with the names.

(2) The authors should explain why they set the antigenicity threshold to 0.5 in the manuscript. Do they have a reference for this value?

Author’s Response

We are thankful for reviewer comments. As per recommendations of reviewer’s, we have modified the manuscript and additional information have been highlighted in the manuscript at page no. 24 and line no. 543-545. Reference is added as per recommendations.

(3) Please calculate and report the aliphatic index and GRAVY parameters of the vaccine construct.

Author’s Response

We are thankful for reviewer comments. As per reviewer’s recommendations, we have revised the manuscript and changes have been highlighted in the manuscript at page no.19 and line no. 414 and 415.

(4) The authors should perform disulfide engineering of the vaccine construct.

Author’s Response

We are thankful for reviewer comments. As per recommendations of reviewer’s, we have performed the Disulfide engineering of our polyvalent construct and changes have been highlighted in the manuscript at page no. 9 and 20 at line no. 203-206 and 433-437 respectively. Figure 10 and Table 6 was added in result section.

(5) Please compare the characteristics of the vaccines designed in this study with the vaccines designed in other authors' studies. For this, please use the following references and cite them.

Author’s Response

We are thankful for reviewer comments. As per recommendations of reviewer’s, we have compared the characteristics of the provided paper links at page no.8 and 19 at line no. 175-177, 412 and 417 respectively also cited them in discussion portion.

Reviewer 2

(1) The manuscript is a bioinformatic analysis of TSA and TAA for Liver cancer. Although the immunoinformatic analysis, molecular dynamics analysis, immune simulation studies are also sufficient to predict the usage of this epitope-based vaccine but unless it is validated by using cancer cell lines or patients’ cells, the results cannot be confirmed. Animal model studies with immunogenic response and tumour regression is required to further validate their claim. So many such vaccine candidates are predicted for almost all the diseases and cancers specially in COVID times and are still being predicted, but they need to be validated preclinically or experimentally.

Author’s Response

 We are thankful for reviewer comments. This is in silico-based research paper on designing and in silico screening analysis of peptides against liver cancer. These peptides are under experimental trials on rat models.

Syed Aun Muhammad, PhD

Corresponding Author

---

## [Decision Letter · Decision Letter 1]

6 Aug 2024

Molecular Dynamics Simulation based Prediction of T-cell Epitopes for the Production of Effector Molecules for Liver cancer immunotherapy

PONE-D-24-20487R1

Dear Dr. Muhammad,

We’re pleased to inform you that your manuscript has been judged scientifically suitable for publication and will be formally accepted for publication once it meets all outstanding technical requirements.

Kind regards,

Sheikh Arslan Sehgal, PhD

Academic Editor

PLOS ONE

Additional Editor Comments (optional):

Reviewers' comments:

Reviewer's Responses to Questions

**Comments to the Author**

1. If the authors have adequately addressed your comments raised in a previous round of review and you feel that this manuscript is now acceptable for publication, you may indicate that here to bypass the “Comments to the Author” section, enter your conflict of interest statement in the “Confidential to Editor” section, and submit your "Accept" recommendation.

Reviewer #1: All comments have been addressed

2. Is the manuscript technically sound, and do the data support the conclusions?

Reviewer #1: Yes

3. Has the statistical analysis been performed appropriately and rigorously? 

Reviewer #1: N/A

4. Have the authors made all data underlying the findings in their manuscript fully available?

Reviewer #1: Yes

5. Is the manuscript presented in an intelligible fashion and written in standard English?

Reviewer #1: Yes

6. Review Comments to the Author

Reviewer #1: -Dear authors, I would like to thank you for carefully considering all the comments. I hope that this manuscript can have a great impact on liver cancer immunotherapy after its final acceptance.

7. PLOS authors have the option to publish the peer review history of their article (what does this mean?). If published, this will include your full peer review and any attached files.

Reviewer #1: No

---

## [Editor Report · Acceptance letter]

8 Aug 2024

PONE-D-24-20487R1 

PLOS ONE

Dear Dr. Muhammad, 

I'm pleased to inform you that your manuscript has been deemed suitable for publication in PLOS ONE. Congratulations! Your manuscript is now being handed over to our production team.

Kind regards, 

on behalf of

Dr Sheikh Arslan Sehgal 

Academic Editor

PLOS ONE